# ANATOMY OF CATASTROPHIC FORGETTING: HIDDEN REPRESENTATIONS AND TASK SEMANTICS

**Vinay V. Ramasesh**
Blueshift, Alphabet
Mountain View, CA

**Ethan Dyer**
Blueshift, Alphabet
Mountain View, CA

**Maithra Raghu**
Google Brain
Mountain View, CA

## ABSTRACT

Catastrophic forgetting is a recurring challenge to developing versatile deep learning models. Despite its ubiquity, there is limited understanding of its connections to neural network (hidden) representations and task semantics. In this paper, we address this important knowledge gap. Through quantitative analysis of neural representations, we find that deeper layers are disproportionately responsible for forgetting, with sequential training resulting in an erasure of earlier task representational subspaces. Methods to mitigate forgetting stabilize these deeper layers, but show diversity on precise effects, with some increasing feature reuse while others store task representations orthogonally, preventing interference. These insights also enable the development of an analytic argument and empirical picture relating forgetting to task semantic similarity, where we find that maximal forgetting occurs for task sequences with intermediate similarity.

## 1 INTRODUCTION

While the past few years have seen the development of increasingly versatile machine learning systems capable of learning complex tasks (Stokes et al., 2020; Raghu & Schmidt, 2020; Wu et al., 2019b), *catastrophic forgetting* remains a core capability challenge. Catastrophic forgetting is the ubiquitous phenomena where machine learning models trained on non-stationary data distributions suffer performance losses on older data instances. More specifically, if our machine learning model is trained on a sequence of tasks, accuracy on earlier tasks drops significantly. The catastrophic forgetting problem manifests in many sub-domains of machine learning including continual learning (Kirkpatrick et al., 2017), multi-task learning (Kudugunta et al., 2019), standard supervised learning through input distribution shift (Toneva et al., 2019; Snoek et al., 2019; Rabanser et al., 2019; Recht et al., 2019) and data augmentation (Gontijo-Lopes et al., 2020).

Mitigating catastrophic forgetting has been an important research focus (Goodfellow et al., 2013; Kirkpatrick et al., 2017; Lee et al., 2017; Li et al., 2019; Serrà et al., 2018; Ritter et al., 2018; Rolnick et al., 2019), but many methods are only effective in specific settings (Kemker et al., 2018), and progress is hindered by limited understanding of catastrophic forgetting's fundamental properties. How does catastrophic forgetting affect the hidden representations of neural networks? Are earlier tasks forgotten equally across all parameters? Are there underlying principles common across methods to mitigate forgetting? How is catastrophic forgetting affected by (semantic) similarities between sequential tasks? This paper takes steps to answering these questions, specifically:

1. With experiments on split CIFAR-10, a novel distribution-shift CIFAR-100 variant, CelebA and ImageNet we analyze neural network layer representations, finding that higher layers are disproportionately responsible for catastrophic forgetting, the sequential training process erasing earlier task subspaces.
2. We investigate different methods for mitigating forgetting, finding that while all stabilize higher layer representations, some methods encourage greater feature reuse in higher layers, while others store task representations as orthogonal subspaces, preventing interference.
3. We study the connection between forgetting and task semantics, finding that semantic similarity between subsequent tasks consistently controls the degree of forgetting.
4. Informed by the representation results, we construct an analytic model that relates task similarity to representation interference and forgetting. This provides a quantitative empirical measure of

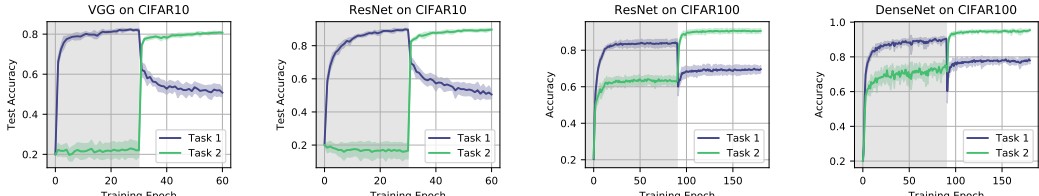

Figure 1: **Examples of catastrophic forgetting across different architectures and datasets.** We plot accuracy of Task 1 (purple) and Task 2 (green), on both the split CIFAR10 task, and the CIFAR100 distribution-shift task across multiple architectures. Catastrophic forgetting is seen as the significant drop of Task 1 accuracy when Task 2 training begins.

task similarity, and together these show that forgetting is most severe for tasks with intermediate similarity.

## 2 RELATED WORK

**Mitigation strategies** Developing mitigation strategies for catastophic forgetting is an active area of research (Kemker et al., 2018). Popular approaches based on structural regularization include Elastic Weight Consolidation (Kirkpatrick et al., 2017) and Synaptic Intelligence (Zenke et al., 2017). An alternative, functional regularization approach is based on storing and replaying earlier data in a replay buffer (Schaul et al., 2015; Robins, 1995; Rolnick et al., 2019). We study how these mitigation methods affect the internal network representations to better understand their role in preventing catastrophic forgetting.

**Understanding catastrophic forgetting** Our work is similar in spirit to existing empirical studies aimed at better understanding the catastrophic forgetting phenomenon (Goodfellow et al., 2013; Toneva et al., 2019). We focus specifically on understanding how layerwise representations change, and on the relation between forgetting and task semantics, which have not previously been explored. This semantic aspect is related to recent work by Nguyen et al. (2019) examining the influence of task sequences on forgetting. It also builds on work showing that learning in both biological and artificial neural networks is affected by semantic properties of the training data (Saxe et al., 2019; Mandler & McDonough, 1993).

**Fine-tuning and transfer learning** While not studied in the context of catastrophic forgetting, layerwise learning dynamics have been investigated in settings other than continual learning. For example, Raghu et al. (2017) showed that layerwise network representations roughly converge *bottom-up*; (from input to output). Neyshabur et al. (2020) observed similar phenomena in a transfer-learning setting with images, as have others recently in transformers for NLP (Wu et al., 2020; Merchant et al., 2020). Furthermore, the observation that early layers in networks learn general features like edge detectors, while latter layers learn more task specific features is a well-known result in computer vision (Erhan et al., 2009), and here we quantitatively study its ramifications for sequential training and catastrophic forgetting.

## 3 SETUP

**Tasks:** We conduct this study over many different tasks and datasets: (i) Split CIFAR-10, where the ten class dataset is split into two tasks of 5 classes each (ii) input distribution shift CIFAR-100, where each task is to distinguish between the CIFAR-100 superclasses, but input data for each task is a different subset of the constituent classes of the superclass (see Appendix A.3) (iii) CelebA attribute prediction: the two tasks have input data either men or women, and we predict either smile or mouth open (iv) ImageNet superclass prediction, similar to CIFAR100.

**Models:** We perform experiments with three common neural network architectures used in image classification — VGG (Simonyan & Zisserman, 2014), ResNet (He et al., 2015) and DenseNet (Huang et al., 2016). Examples of catastrophic forgetting in these models are shown in Figure 1.

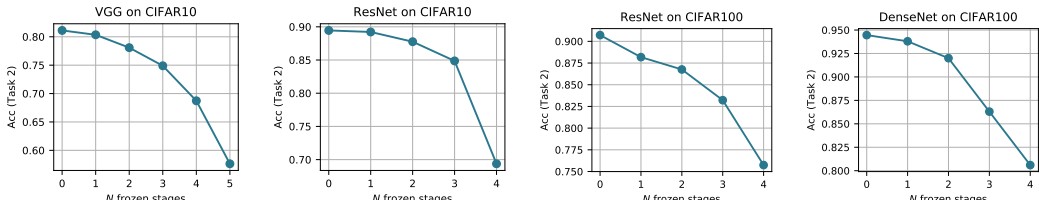

Figure 2: **Freezing lower layer representations after Task 1 training has little impact on Task 2 accuracy.** We freeze the parameters of a contiguous block of layers (starting from the lowest layer) after Task 1 training, and only train the remainder on Task 2, finding that freezing the lowest layers has little impact on Task 2 accuracy.

# 4 CATASTROPHIC FORGETTING AND HIDDEN LAYER REPRESENTATIONS

We begin by investigating how catastrophic forgetting manifests in the hidden representations of the neural network. Do all parameters (and layers) forget equally? Or are specific components of the network particularly responsible for the drop in accuracy in sequential training? Through a combination of layer freezing experiments, and representational analysis, we find that *higher layers* (layers closest to output) are disproportionately responsible for catastrophic forgetting, with *lower layers* (layers closest to input) remaining representationally stable through sequential training. Further analysis of the representational subspaces of Task 1, Task 1 (post Task 2 training) and Task 2, provide insights on both feature reuse and interference.

## 4.1 FREEZING LAYER REPRESENTATIONS

To study the effect of individual layers on forgetting, we measure the effect of freezing layer representations on Task 2 accuracy (Figure 2). Specifically, we freeze a contiguous block of layers (starting from the lowest layer) after training on Task 1, and only train the remaining layers on Task 2. Across different architectures and tasks, we observe that lower layers can be reliably frozen with very little impact on Task 2 accuracy. This suggests the possibility of lower layer features being reused between both tasks, with higher layers being the main contributor to catastrophic forgetting.

## 4.2 REPRESENTATIONAL SIMILARITY THROUGH SEQUENTIAL TRAINING

The results of Figure 2 illustrate that lower layer representations on Task 1 can be reused for good performance on Task 2. To determine if this is what actually occurs during the training process, we turn to Centered Kernel Alignment (CKA) (Kornblith et al., 2019), a neural network representation similarity measure. CKA and other related algorithms (Raghu et al., 2017; Morcos et al., 2018) provide a scalar score (between 0 and 1) determining how similar a pair of (hidden) layer representations are, and have been used to study many properties of deep neural networks (Gotmare et al., 2018; Kudugunta et al., 2019; Wu et al., 2019a).

Specifically, letting $X \in \mathbb{R}^{n \times p}$ and $Y \in \mathbb{R}^{n \times p}$ be (centered) layer activation matrices of (the same) $n$ datapoints and $p$ neurons, CKA computes

$$\text{CKA}(\text{X}, \text{Y}) = \frac{\text{HSIC}(\text{XX}^{\text{T}}, \text{YY}^{\text{T}})}{\sqrt{\text{HSIC}(\text{XX}^{\text{T}}, \text{XX}^{\text{T}})}\sqrt{\text{HSIC}(\text{YY}^{\text{T}}, \text{YY}^{\text{T}})}} \tag{1}$$

for HSIC Hilbert-Schmidt Independence Criterion (Gretton et al., 2005). We use linear-kernel CKA.

In Figure 3, we plot the results of computing CKA on Task 1 layer representations before and after Task 2 training. Across architectures and tasks, we observe that the lower layers have high representation similarity, suggesting that lower layer Task 1 features are reused in Task 2. The higher layers however show significant decreases in representation similarity, suggesting they disproportionately contribute to catastrophic forgetting. These conclusions are further supported by additional layer reset experiments in Appendix Figure 12, which shows that rewinding higher layer parameters from post-Task 2 training to their pre-Task 2 training values significantly improves Task 1 performance.

## 4.3 FEATURE REUSE AND SUBSPACE ERASURE

Further insights on how the representations of lower and higher layers evolve during sequential training is given through a subspace similarity analysis. Letting $X \in \mathbb{R}^{n \times p}$ be the (centered) layer activation matrix of $n$ examples by $p$ neurons, we compute the PCA decomposition of $X$, i.e. the

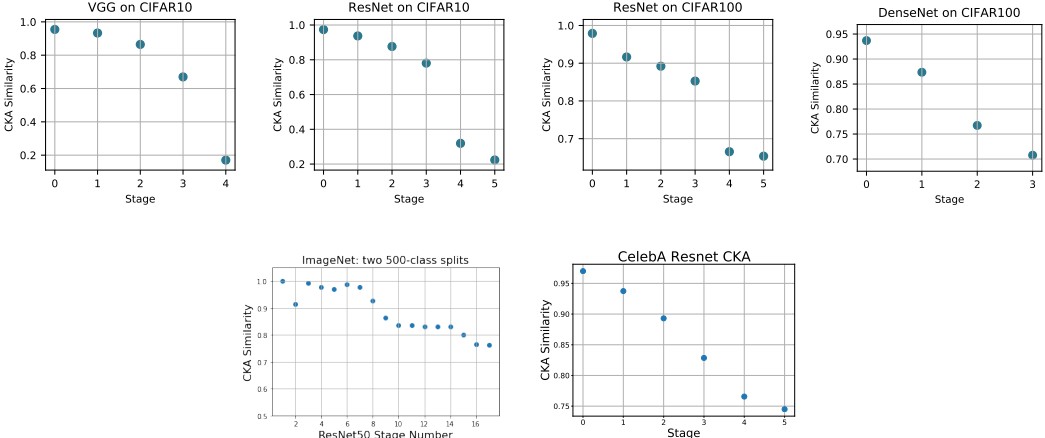

Figure 3: **CKA analysis of layer representations before/after Task 2 training shows lower layer feature reuse and higher layer representations changing significantly on CIFAR10, CIFAR100, ImageNet and CelebA.** We compute CKA of the Task 1 layer representations before and after Task 2 training. We observe that later layers change more than early layers through second task training.

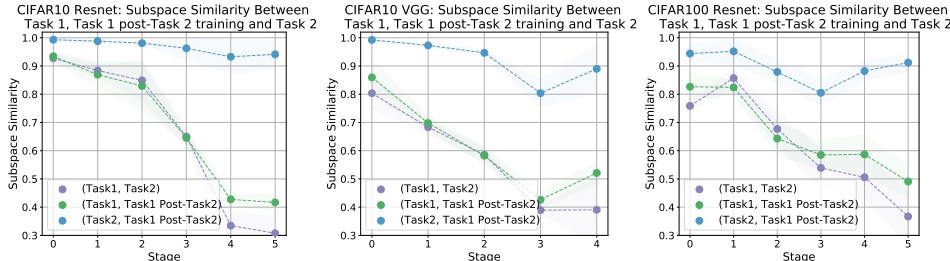

Figure 4: **Subspace similarity analysis shows feature reuse in lower layers and Task 1 subspace erasure in higher layers.** For Resnet on CIFAR10 (left), VGG on CIFAR10 (middle), Resnet on CIFAR100 (right), we compute subspace similarity between: (i) (Task 1, Task 2) (purple line) (ii) (Task 1, Task 1 post-Task 2 training) (green line) (iii) (Task 2, Task 1 post-Task 2 training) (blue line). All comparisons show high similarity in lower layers, indicative of feature reuse. Most striking is the comparison between (Task 1, Task 1 post-Task 2 training) and (Task 2, Task 1 post-Task 2 training), which show that after Task 2 training, higher layer representations for Task 1 are more similar to Task 2 than to Task 1! Specifically, the Task 1 subspace in higher layers is erased through Task 2 training.

eigenvectors $(v_1, v_2, ...)$ and eigenvalues $(\lambda_1, \lambda_2...)$ of $X^T X$. Letting $V_k$ be the matrix formed from the top $k$ principal directions, $v_1, ..., v_k$ as columns, and $U_k$ the corresponding matrix for a different activation matrix $Y$, we compute

$$\text{SubspaceSim}_k(X, Y) = \frac{1}{k} ||V_k^T U_k||_F^2 .$$

This measures the overlap in the subspaces spanned by $(v_1, ..., v_k)$ and $(u_1, ..., u_k)$. Concretely, if $X$ and $Y$ correspond to layer activation matrices for two different tasks, $\text{SubspaceSim}_k(X, Y)$ measures how similarly the top $k$ representations for those tasks are *stored* in the network.

Figure 4 shows the result of computing $\text{SubspaceSim}_k(X, Y)$ for $X, Y$ being the layer activation matrices in (i) (Task 1, Task 2) (purple line) (ii) (Task 1, Task 1 post-Task 2 training) (blue line) (iii) (Task 2, Task 1 post-Task 2 training) (blue line). Supporting the CKA analysis, lower layers have high subspace similarity between Task 1 and Task 2 (feature reuse), but higher layers have low similarity (representations change significantly during Task 2 training.) Furthermore, (Task 1, Task 1 post-Task 2 training) also shows low subspace similarity in the higher layers, indicating that the representational change in later layers is not restricted to Task 2 data, but also alters the networks representation of the initial task. Strikingly, (Task 2, Task 1 post-Task 2 training) shows high similarity in all layers, despite being *different tasks*.

In summary, these results illustrate that during sequential training, effective feature reuse happens in the lower layers, but in the higher layers, after Task 2 training, Task 1 representations are mapped

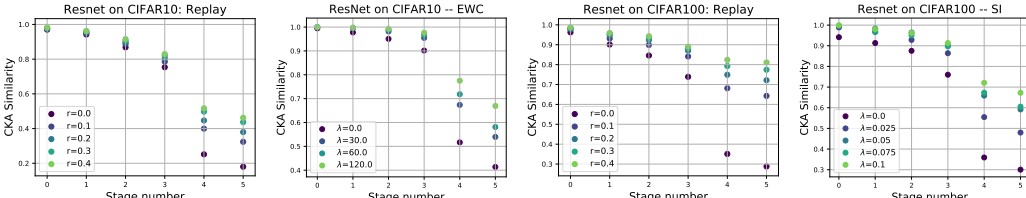

Figure 5: **CKA analysis of mitigation methods demonstrates that mitigation methods stabilize the higher layer representations.** We compute CKA between layer representations of Task 1 before and after Task 2 training, with varying amounts and types of mitigation. We find that across all mitigation methods, even a small amount of mitigation works to stabilize the higher layer representations. In Figure 6, we investigate whether this stabilization results in greater feature reuse or through finding independent subspaces.

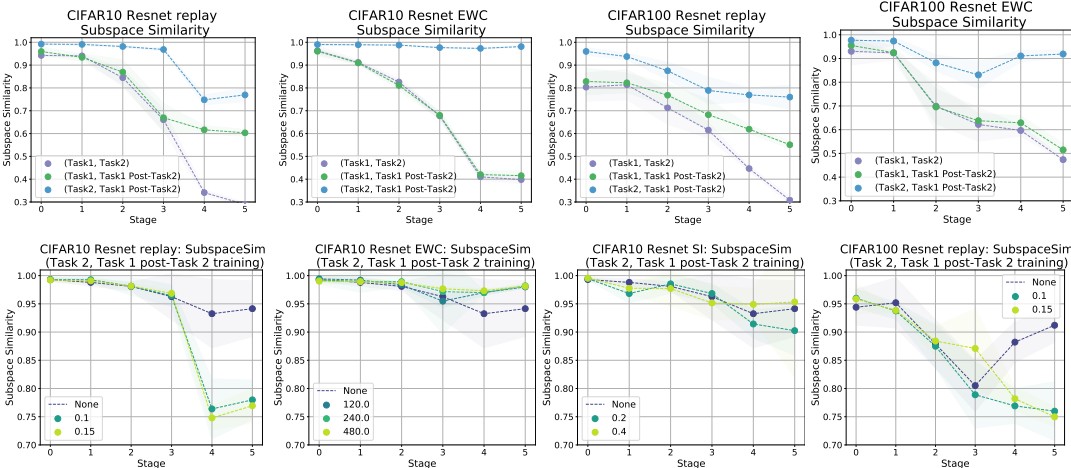

Figure 6: **Subspace similarity analysis reveals differences between different mitigation methods, with replay storing Task 1 and Task 2 representations in orthogonal subspaces, while EWC and SI promote feature reuse in the higher layers.** Top row: We plot subspace similarities like in Figure 4, for different mitigation methods. We observe that (Task 2, Task 1 post-Task 2 training) is much lower in replay compared to EWC/SI, as is (Task1, Task2) similarity. The bottom row plots (Task 2, Task 1 post-Task 2 training) for different amounts of mitigation. Again we see that replay has much lower (Task 2, Task 1 post-Task 2 training) similarity than no mitigation, while EWC/SI maintain the similar subspaces of Task 1 and Task 2 post Task 2 training.

into the *same* subspace as Task 2. Specifically, Task 2 training causes *subspace erasure* of Task 1 in the higher layers. (Additional details and results are in Appendix Section A.5.)

## 5 FORGETTING MITIGATION METHODS AND REPRESENTATIONS

Informed by this understanding of how catastrophic forgetting manifests in the hidden layers (feature reuse in lower layers and subspace erasure in higher layers), we next study methods to mitigate forgetting. Do popularly used mitigation methods act to stabilize higher layers — given possible alternate approaches such as weight orthogonalization (Appendix C)?

We investigate these questions across different successful mitigation strategies (i) EWC (elastic weight consolidation) (Kirkpatrick et al., 2017), a regularization-based approach (ii) replay buffer (Schaul et al., 2015; Ratcliff, 1990; Rolnick et al., 2019), a rehearsal-styled approach (iii) SI (Synaptic Intelligence) (Zenke et al., 2017), another popular regularization-based approach.

In Figure 5, we plot the results of computing CKA on layer representations of Task 1 before/after training on Task 2, with different amounts of mitigation applied, and across different mitigation methods, tasks and architectures (full results in Appendix B.7.) Across all of these settings, we observe that even a small amount of mitigation results in a significant increase in similarity (stabilization) in the higher layers. However, there remains a central open question — is this increase in similarity due to *feature reuse in the higher layers* between Task 1 and Task 2, or does the network store Task 1 and Task 2 representations in non-interfering, *orthogonal subspaces*?

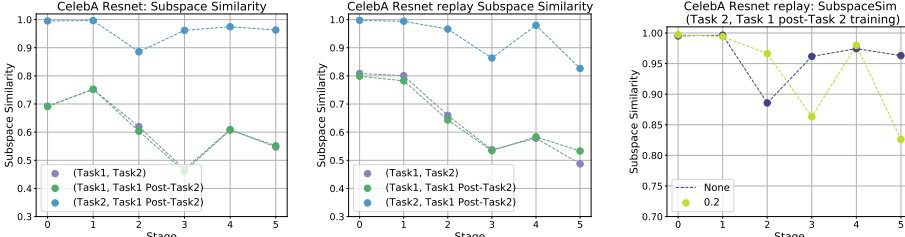

Figure 7: **Subspace similarity analysis of CelebA, with and without replay mitigation.** Left pane computes subspace similarities in CelebA, like in Figure 4, showing subspace erasure at top layers. Middle and right panes showt the effect of adding replay, where subspaces in lower/middle layers become more similar but higher layers become more orthogonal, like in Figure 6.

To answer this, we compute the Subspace Similarity measure introduced in Section 4.3, with results in Figure 6 (and additional experiments in Appendix Section B.5). These results reveal key differences between the mitigation methods: replay buffer results in orthogonal network subspaces for higher layer representations of Task 1 and Task 2, while EWC and SI promote feature reuse even in the higher layers. In more detail:

- Comparing the top row of Figure 6 to Figure 4 (the same plot without mitigation), we see that mitigation methods increase the subspace similarity of (Task 1, Task 1 Post-Task 2) and (Task 1, Task 2) representations in earlier/middling layers (more feature reuse).
- Most interestingly, comparing the different mitigation methods across top row Figure 6, we see that while EWC and SI maintain high subspace similarity of (Task 2, Task 1 Post-Task 2) even in the highest layers, replay significantly decreases this value in the highest layers. This suggests that replay performs mitigation through use of orthogonal subspaces, while EWC/SI encourage feature reuse even in the highest layers.
- In the bottom row we show subspace similarity of (Task 2, Task 1 Post-Task 2) for varying strengths of the different mitigation methods. We observe a clear decrease in subspace similarity in higher layers even when a little replay is used (again supporting orthogonal subspaces), while EWC/SI maintain or increase this (feature reuse even in higher layers.)

## 6 SEMANTICS

With the insights on how catastrophic forgetting relates to the hidden representations, and the effects on network internals of different mitigation methods, we turn to investigating how the semantic similarity between the sequential tasks affects forgetting. Specifically the prior results on lower layer feature reuse suggest a key question: does similarity between tasks result in less forgetting? Surprisingly, the answer is quite nuanced, depending on how precisely the different tasks are represented in the network.

### 6.1 FORGETTING, TASK SEMANTICS AND A SIMILARITY PUZZLE

Our study of the relationship between task semantic similarity and forgetting reveals a surprising puzzle. In Figure 8 and Table 1, we observe that depending on the task setup, a semantically similar task may be forgotten less *or* more than a less similar task. This behavior is consistent across different settings and datasets. Specifically, in Figure 8a and 8d, Task 1 and Task 2 consist of binary classifications between animals and objects from CIFAR-10, where we find that the similar categories forget less. But in Figure 8b and 8e, a four category classification between animals and objects followed by a binary classification task results in the similar categories being forgotten more. Table 1 shows an analogous result on ImageNet. This effect of similar categories being forgotten more can also be observed in the CIFAR-100 distribution shift setting, where input distribution shift to the CIFAR-100 superclasses results in more similar categories being forgotten more, e.g. input shifts to natural superclasses (Figure 8c) or artifical superclasses (Figure 8f). Further details and supporting experiments can be found in Appendix B.9.

In the subsequent sections, we resolve this puzzle through (i) the development of an analytic model that formalizes the subtle connection between task similarity and forgetting (ii) an ensuing measure of

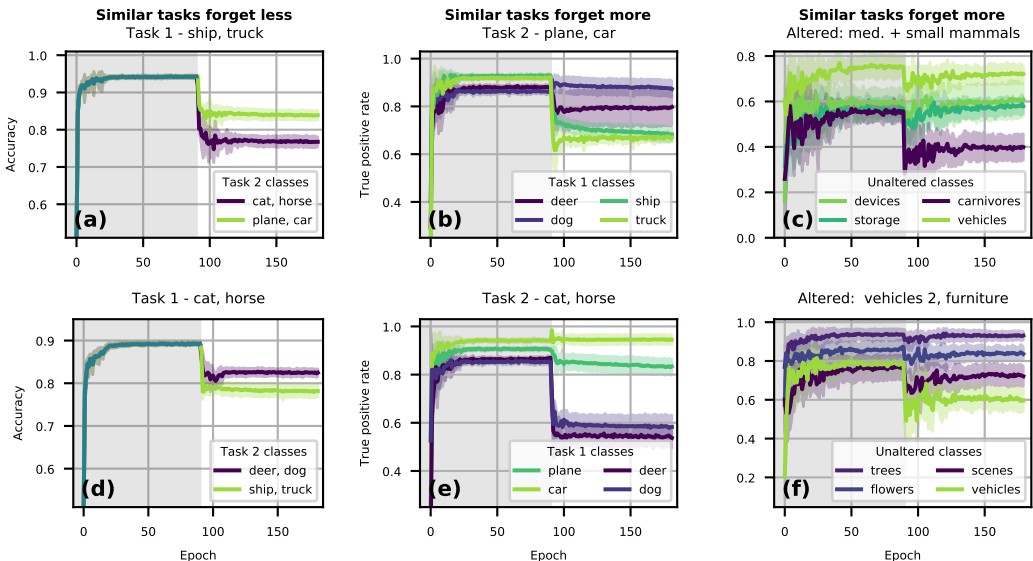

Figure 8: **Forgetting severity consistently varies along semantic divisions.** (a,d) For sequential binary classification tasks models trained on either objects or animals forgets less on a second object task of the same semantic category. (b,e) Models trained initially on four way classification of two object and two animal classes see more forgetting in semantically similar classes when trained on a second two class task of either objects or animals. (c,f) Models trained on sequential six-way classification with four unaltered categories (shown) and two altered categories show increased forgetting in the unaltered categories semantically most similar to altered categories. For all panes, performance is shown for ResNet, with additional results in Appendix B.9.

| Task Description | Performance Drop on Task 1 | |
| --- | --- | --- |
| | Animals | Artifacts |
| Task1: animals Task2: animals | **0.0885 ± 0.0259** | - |
| Task1: animals Task2: artifacts | 0.1832 ± 0.0484 | - |
| Task1: artifacts Task2: artifacts | - | **0.2884 ± 0.0187** |
| Task1: artifacts Task2: animals | - | 0.3720 ± 0.0288 |
| Task1: animals and artifacts Task2: animals | **0.1752 ± 0.0193** | 0.1532 ± 0.0231 |
| Task1: animals and artifacts Task2: artifacts | 0.2057 ± 0.0312 | **0.2330 ± 0.0292** |

Table 1: **Task Semantics and forgetting in ImageNet** Here a ResNet50 is first trained (task 1) to classify either 10 animals, 10 artifacts or 10 animals and artifacts (chosen randomly from ImageNet). The network is then trained (task 2) to either classify 10 different animals or 10 different artifacts. The table shows performance drop on Task 1 after Task 2 training. Similar to Figure 8, when Task 1 is either animals or artifacts (top part of table), similar tasks forget less, while when Task 1 is both animals and artifacts, similar tasks show greater forgetting (bottom part of table).

task similarity, *trace overlap*, empirically quantifies task similarity (iii) showing maximal forgetting happens with intermediate task similarity.

## 6.2 AN ANALYTIC MODEL OF FORGETTING SEMANTICS

We first formalize a model with a single classification head (like in distribution shift CIFAR-100), with the corresponding multi-head setup in Appendix C. We write the network output as $f(x) = \sum_{\mu=1}^{p} \theta_\mu g_\mu(w; x)$, where $\theta$ are the last layer weights and $g$ are features. We consider training sequentially on two tasks. For the first task, we train normally. For the second task, inspired by our empirical observation above that forgetting is driven by deeper layers, we freeze the the earlier layer features, $g_\mu(w; x) = g_\mu(\hat{w}; x)$ when training on the second task (here $\hat{w}$ represent the weights after training on the first task). If we train with loss function $L(f, y)$ and learning rate $\eta$, the network output SGD update at timestep $t$ is

$$\Delta f_t(x) = -\eta \sum_{x', y' \in \mathcal{D}_{\text{train}}^{(2)}} \Theta(x, x') \frac{\partial L(f(x'), y')}{\partial f} . \tag{2}$$

Here $\mathcal{D}_{\text{train}}^{(2)}$ is the second-task training data and $\Theta(x, x') = \sum_{\mu=1}^{p} g_\mu(\hat{w}; x) g_\mu(\hat{w}; x')$ is the *overlap* (inner product) between the model last-layer representations on data point $x$ and $x'$. If the overlap $\Theta(x, x')$ is small, then the change in the logits is small and forgetting is minimal, specifically:

**Lemma 1** *Let $f_t$ be the logits of a neural network trained via the frozen feature model. Let $x$ be an element of the initial task test set. Let us further denote the vector of the feature overlap applied to the second task training set as $\vec{\Theta}(x) := \{\Theta(x, x') : x' \in X_{train}^{(2)}\}$ and the loss vector as $\vec{L} = \{L(f(x'), y') : x', y' \in \mathcal{D}_{train}^{(2)}\}$. With this, the change in the original task logits is bounded as*

$$|\Delta f_t(x)| \leq \eta \left\| \vec{\Theta}(x) \right\| \left\| \frac{\partial \vec{L}}{\partial f} \right\| . \tag{3}$$

Thus representational similarity is necessary for forgetting to occur, and in this model, sufficiently similar and sufficiently dissimilar tasks have minimal forgetting, while intermediate similar tasks have maximal forgetting.

## 6.3 CATASTROPHIC FORGETTING ACROSS VARYING TASK SIMILARITIES

The analytical result enables us to make precise a measure of task similarity, through using the feature overlap matrix $\Theta(x', x)$. Concretely, letting $\Theta_{11}$ be the overlap matrix on Task 1, $\Theta_{22}$ on Task 2, and $\Theta_{12}$ on Task 1 and Task 2, we define:

$$\text{Trace overlap} = \frac{\text{Tr}\left(\Theta_{12}\Theta_{12}^T\right)}{\sqrt{\left(\text{Tr}\left(\Theta_{11}\Theta_{11}^T\right) \text{Tr}\left(\Theta_{22}\Theta_{22}^T\right)\right)}} . \tag{4}$$

Trace overlap is directly analogous to CKA, where instead of comparing the similarity of different features on the same data, we are comparing the similarity of common features on different data.

**Increasing task *dissimilarity* can help reduce forgetting** In Lemma 1, we see that very dissimilar tasks, with small overlap matrix, exhibit minimal forgetting. In Figure 9, we test this in the context of sequential binary classification. We introduce an *other* category to the classification problem (of images from classes not included in the training tasks), which encourages the model to represent Task 1 and Task 2 dissimilarly (lower trace overlap in Figure 9b), and reduces forgetting (Figure 9a).

**Maximal forgetting at intermediate similarity** To empirically study the conclusion from the model that intermediate-similar tasks have maximal forgetting, we use mixup (Zhang et al., 2017), a standard data augmentation technique, to interpolate between the two tasks. Specifically, we first train a model on binary classification (with an other category to encourage dissimilarity) and for $\lambda \in [0, 1]$, generate an mixup dataset $\mathcal{D}_{\text{train}}^{(\lambda)} = \{(\lambda x^{(2)} + (1-\lambda)x^{(1)}, \lambda y^{(2)} + (1-\lambda)y^{(1)})\}$. We compare model behavior across a range of mixup fractions, finding that the task similarity as measured through the trace

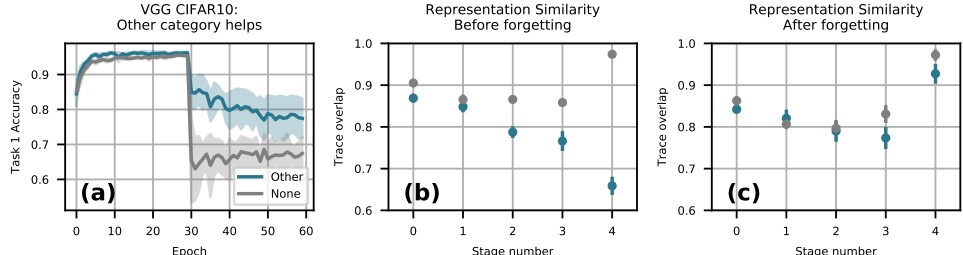

Figure 9: **Adding an *other* category reduces forgetting and increases orthogonalization**. (a) Models trained initially on classifying two objects: airplane/automobile, and subsequently on two animals: deer/dog exhibit diminished forgetting when we add an *other* category to task 1 training. The other category contains randomly sampled images from all classes not present in the initial or final tasks. (b,c) The presence of the other category reduces the similarity between the model's representations of each task. Additional plots are in Appendix B.8.

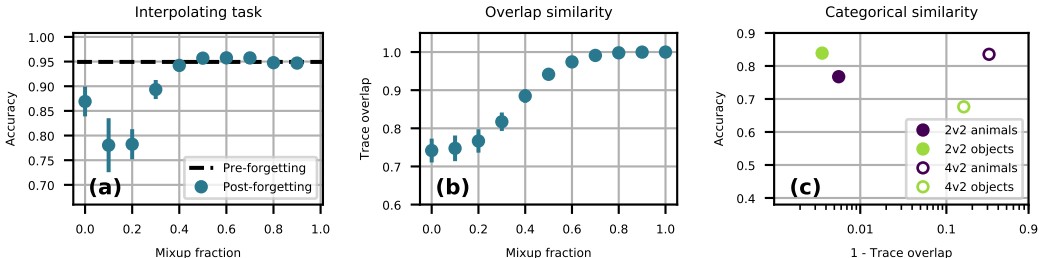

Figure 10: **Task similarity affects performance.** (a) We measure initial task performance for sequential binary classification as we dial the mixup fraction between 0.0, pure task 2 data, and 1.0, pure task 1 data. Maximum forgetting (minimal accuracy) occurs at an intermediate value of the mixup fraction, $\sim 0.1 - 0.2$. (b) We measure the similarity between task 1 and task 2 as we vary the mixup fraction, supporting the intuition that interpolating mixup fraction interpolates task similarity. (c) Task 1 performance after task 2 training is plotted against task 1 versus task 2 representational similarity for animal and object classes in the sequential binary classification tasks (solid) and four-way followed by binary classification tasks (hollow) of Figures 8a and Figures 8b. In the case of sequential binary tasks both object and animal second tasks are represented extremely similarly to the initial object task and the slightly less similar, object task, forgets more. For the four way classification task, animals are represented dissimilarly to the second object task and cause less forgetting.

overlap between task1 and task2 varies monotonically with mixup fraction (Figure 10b), quantifying that varying the mixing parameter, $\lambda$, tunes the degree of task similarity. Figure 10a shows that the most extreme forgetting happens for intermediate similarity (i.e. mixing fraction $\sim 0.1 - 0.2$).

**Semantic similarity revisited** Armed with these quantitative measures, we revisit the task semantics puzzle in Figure 8 where (i) for sequential binary classification, semantically more similar categories were forgotten less, while in (ii) four class classification followed by binary classification and input distribution shift, similar classes were forgotten more. Figure 10c (solid dots) shows that in (i) the net didn't represent objects and animals differently, and the slightly more similar task is forgotten less. But in (ii) Figure 10c (hollow dots), the model encodes animals and objects differently, minimizing forgetting on the dissimilar tasks, consistent with maximal forgetting for intermediate similarity.

## 7 CONCLUSION

In this paper, we have studied some of the fundamental properties of catastrophic forgetting, answering important open questions on how it arises and interacts with hidden representations, mitigation strategies and task semantics. By using representational analysis techniques, we demonstrated that forgetting is not evenly distributed throughout the deep learning model but concentrated at the higher layers, which change significantly and erase earlier task subspaces through sequential training. Consistent with this, we find mitigation methods all stabilize higher layer representations, but vary on whether they enforce more feature reuse, or store tasks in orthogonal subspaces. Informed by these insights, we formulate an analytic model to investigate connections between forgetting and task

semantics. This gives a quantitative measure of task similarity and shows that intermediate similarity in sequential tasks leads to maximal forgetting. In summary, these results help provide a foundation for a deeper understanding of catastrophic forgetting and suggest new approaches for developing and measuring mitigation methods.

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

# A    METHODS

## A.1    MODELS

Here we detail the model architectures used in this study. We use standard versions of the popular VGG (Simonyan & Zisserman, 2014), ResNet (He et al., 2015), and DenseNet (Huang et al., 2016) architectures. These architectures have varying structural properties (e.g., presence or absence of skip connections), meaning observed behaviors common to all of them are likely to be robust and generalizable.

**VGG**: The VGG model we use consists of five stages. Each stage comprises a convolutional layer, a ReLU nonlinearity, another convolutional layer, another ReLU nonlinearity, and finally a MaxPool layer. Each convolution uses a 3-by-3 kernel with unit stride and padding. The MaxPool operation uses a 2-by-2 kernel with stride 2. The number of channels by stage is 16, 32, 64, 128, 128. We do not use batch normalization in the VGG.

We test two different version of the VGG: one with just the stages described above, and another with two fully connected layers, of width 1024, after the convolutional layers.

**ResNet**: Our ResNet consists of a initial 3-by-3 convolutional layer with 32 channels, followed by a 2d batch norm operation. This initial pair is followed by four stages, each consisting of two residual blocks per stage. Each block consists of two conv-BN-ReLU sequences, with a shortcut layer directly routing the input to add to the final ReLU preactivations. All convolutions are 3-by-3, with unit padding. In all but the first block, the first convolutional layer downsamples, with a stride of 2; the second convolutional layer maintains stride of 1. The number of channels doubles each block, starting from a base level of 32 channels in the first block.

**DenseNet**: Like ResNet, our DenseNet consists of an initial 3-by-3 convolution, followed by four dense blocks. In between each pair of dense blocks is a transition block. Following the final dense block is a batch-normalization, ReLU, and average pooling operation to generate the final features. Our DenseNet is characterized by a growth rate of 12 and compression rate of 0.5. The first stage of the DenseNet features 6 blocks, with the number of blocks doubling in each subsequent stage.

## A.2    TRAINING

All networks are trained using cross-entropy loss with SGD with momentum ($\beta = 0.9$), using a batch size 128. We do not use learning-rate schedules here, leaving that investigation to future work. To better correspond with practical situations, however, we choose a learning rate and total number of epochs such that interpolation (of the training set) occurs. For the split CIFAR-10 task, this typically corresponds to training for 30 epochs per task with a learning rate of 0.01 (VGG), 0.03 (ResNet), and 0.03 (DenseNet). We use weight decay with strength 1e-4, and do not apply any data augmentation. For the split CIFAR-100 task, we usually train for 60 or 90 epochs per task, with other the other hyperparameters identical to the CIFAR-10 case.

For multi-head networks (used in split CIFAR-10), each head is initialized with weights drawn from a normal distribution with variance $1/n_f$, where $n_f$ is the number of features; biases are initialized to zero. We do not copy the parameters from the old head to the new when switching tasks.

For the split CIFAR-10 setup, the plots shown in the main text refer to experiments in which the initial task was classifying between the five categories *airplane*, *automobile*, *bird*, *cat*, and *deer*; and the second task comprising *dog*, *frog*, *horse*, *ship*, and *truck*. To verify that our results were not unique to the particular split of CIFAR-10 we chose, we used three other splits: *airplane*, *bird*, *deer*, *frog*, *ship* (task 1) and *automobile*, *cat*, *dog*, *horse*, *truck* (task 2); *automobile*, *cat*, *dog*, *horse*, *truck* (task 1) and *airplane*, *bird*, *deer*, *frog*, *ship* (task 2); and *dog*, *frog*, *horse*, *ship*, *truck* (task 1) and *airplane*, *automobile*, *bird*, *cat*, *deer* (task 2).

## A.3    CIFAR-100 DISTRIBUTION SHIFT TASK

Here we provide the concrete example of the CIFAR-100 distribution-shift task shown in Figure 1. This is meant to capture it the scenario where catastrophic forgetting arises through *input distribution shift* — the input data to the neural network undergoes a distribution shift, causing the neural network

to perform poorly on earlier data distributions (Arivazhagan et al., 2019; Snoek et al., 2019; Rabanser et al., 2019). As mentioned in the main text, in this task, the model must identify, for CIFAR-100 images, which of the 20 superclasses the image belongs to. The difference in tasks comes from the difference in subclasses which make up the superclass. As an example, we take the five superclasses *aquatic mammals*, *fruits and vegetables*, *household electrical devices*, *trees*, and *vehicles-1*[1], with the corresponding task 1 subclasses (1) dolphin, (2) apple, (3) lamp, (4) maple tree, and (5) bicycle and task 2 subclasses (1) whale, (2) orange, (3) television, (4) willow, and (5) motorcycle. A key feature of this setting is that task-specific components (including multiple heads) are precluded since the model does not know the task identity either at inference or time. While we do not explore it here, this setup also allows for continuously varying the data distribution, another situation likely to occur in practice.

### A.4    REPRESENTATIONAL SIMILARITY AND CENTERED KERNEL ALIGNMENT

To understand properties of neural network hidden representations, we turn to representational similarity algorithms. A key challenge in analyzing neural network hidden representations is the lack of *alignment* — no natural correspondence exists between hidden neurons across different neural networks. Representational similarity algorithms propose different ways to overcome this — one of the first such algorithms, SVCCA (Raghu et al., 2017; Morcos et al., 2018), uses a Canonical Correlation Analysis (CCA) step to align neurons (enable invariance) through linear transformations.

We use centered kernel alignment (CKA), proposed by Kornblith et al. (2019), to measure the similarity between two representations of the same dataset which is invariant to orthogonal transformation and isotropic scaling (but not arbitrary linear transformations). Given a dataset of $m$ examples, we compare two representations $X$ and $Y$ of that dataset (say, from two different neural networks or from the same neural network with different parameters), with $n_x$ and $n_y$ features respectively; that is, $X \in \mathbb{R}^{m \times n_x}$ and $Y \in \mathbb{R}^{m \times n_y}$. Then the linear-kernel CKA similarity between the two representations is given by

$$\text{CKA}(X, Y) = \frac{||X^T Y||_F^2}{||X^T X||_F^2 ||Y^T Y||_F^2} \tag{5}$$

### A.5    SUBSPACE SIMILARITY

For the subspace similarity computations we pick a fixed threshold $k$ for all layers so that the first $k$ principal components capture $80\%$ of variance in the final layer. This gives $k = 3$ for Resnet on CIFAR-10, $k = 4$ for Resnet on CIFAR-100, $k = 4$ for VGG on CIFAR-10 and $k = 7$ for VGG on CIFAR-100 and $k = 4$ for DenseNet on CIFAR-10 and $k = 5$ for DenseNet on CIFAR-100.

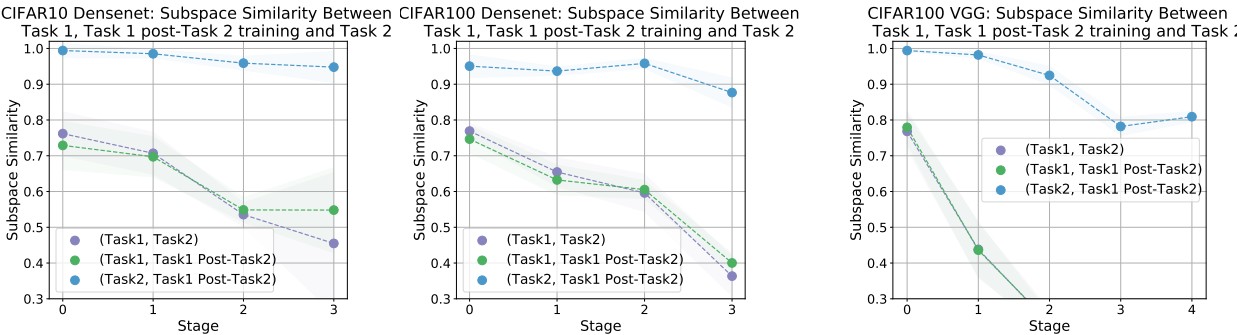

Figure 11: **Subspace similarity analysis shows feature reuse in lower layers and Task 1 subspace erasure in higher layers.**

---

[1]The CIFAR-100 dataset features two vehicle superclasses, denoted *vehicles-1* and *vehicles-2*

### A.6 Mitigation Methods

#### A.6.1 Elastic Weight Consolidation

Developed by Kirkpatrick et al. in 2017, elastic weight consolidation (EWC) adds a term to the loss function of the new task to get the regularized loss function $\mathcal{L}(\vec{\theta})$:

$$\mathcal{L}(\vec{\theta}) = \mathcal{L}_B(\vec{\theta}) + \frac{\lambda}{2} \sum_i F_i \cdot \left(\theta_i - \theta_{A,i}^*\right)^2 . \tag{6}$$

Here, $\vec{\theta}$ are the model parameters; $\mathcal{L}_B(\vec{\theta})$ is the unregularized loss function of the new task; $\lambda$ is a parameter which controls the strength of the regularization; $F_i$ is the diagonal of the Fisher information matrix with respect to the parameters; and $\vec{\theta}_A^*$ are the model parameters after having been trained on task A. We compute the Fisher information via using the squared gradients of the log-likelihood, averaged over a subset of the trainset. We use the empirical Fisher as an approximation.

Following standard practice, in our experiments we estimate the Fisher information using a subset of the full training data; for all of our models on CIFAR-10, 200 samples proved sufficient to be reasonably confident of a converged estimate. We do not include a delay between the start of second-task training and the application of the EWC penalty. Naturally, we do not apply the EWC penalty to the parameters of the head in the multi-head setting.

#### A.6.2 Synaptic Intelligence

Developed by Zenke et al. (2017), synaptic intelligence, like EWC, is a structural regularization method which adds an elastic penalty to the second task's loss function:

$$\mathcal{L}(\vec{\theta}) = \mathcal{L}_B(\vec{\theta}) + \frac{\lambda}{2} \sum_i C_i \cdot \left(\theta_i - \theta_{A,i}^*\right)^2 . \tag{7}$$

The difference from EWC is in how the elastic penalties are calculated. In synaptic intelligence, for each parameter, a contribution to the loss is tracked via the product of the gradient of the loss and the stepwise change in the parameter value. The coefficient is then constructed such that changing the parameter by an amount equal to its change during task-1 training will incur a loss penalty equal to the change in loss from the initial value. Our implementation does not differ from the standard implementation.

## B Supporting experiments

Here we present experiments omitted from the main text for space, which help flesh out some of the conclusions. This section is roughly organized with layerwise experiments first, followed by semantic experiments.

### B.1 Layer reset experiments

As described in the main text, we perform an additional experiment to check that deeper representations get corrupted more than shallower representations. This experiment, corresponding to Figure 12, shows that when layers are reset to their pre-forgetting value, performance improves more dramatically when deeper layers are reset first.

### B.2 Task-specific stages

Our observations that forgetting is driven primarily by the higher layers suggests that in a setting with known task identities (i.e. in the split CIFAR-10 setting described in the main text but not the CIFAR-100 distribution-shift setting), making the deeper stages of the network task-specific would likely mitigate a large portion of forgetting on the old tasks while still allowing for full performance on the new tasks. In particular, since the changes in deeper represnations are much greater than their shallower counterparts, only a few task-specific stages should be necessary to get significant performance increases on the old task. For each of the three network architectures we considered

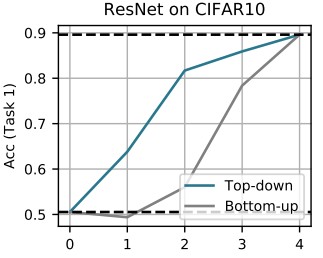 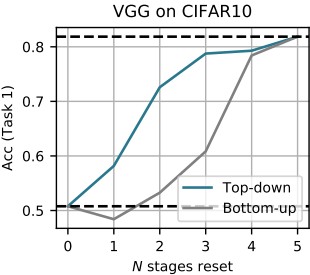 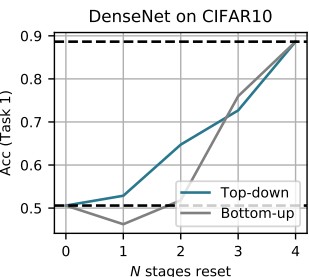

Figure 12: **Layer reset experiments.** After training on Task 2 we reset contiguous blocks of layers to their values before training and record the resulting accuracy on Task 1 (bottom row). We see a significant increase in accuracy when resetting the highest $N$ layers (blue line) compared to resetting the $N$ lowest layers (gray line). Together, these results demonstrate that higher layers are disproportionately responsible for forgetting.

(VGG, ResNet, and DenseNet), we investigated the performance of these task-specific networks on the split CIFAR task. We intend these results to be further evidence of the role of deeper layers in forgetting, not a proposed mitigation scheme.

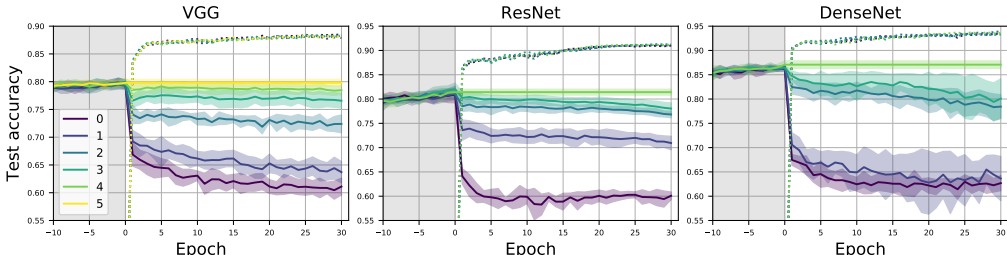

Figure 13: **Forgetting in networks with task-specific stages on CIFAR-10**. Using the same settings as Figure 1 of the main text, we run networks with task-specific layers on split CIFAR-10. We make deeper layers task-specific before shallower layers. Performance increases with number of task-specific layers, denoted by the line color (see legend). In all architectures, two task-specific stages is sufficient to recover significant performance.

Results, shown in figure 13, show that for all architectures, performance dramatically rises with the number of task-specific stages. For all architectures, having the deepest two stages be task-specific recovers a significant fraction of the forgotten task performance.

### B.3 LAYER RESET AND RETRAIN EXPERIMENTS

As an additional probe of the degree to which catastrophic forgetting impacts the representations of various network layers, we perform a reset-and-retrain experiment on the split CIFAR-10 task, using the same training setup as in Figure 3 of the main text. The reset-and-retrain experiment is designed to probe the degree to which network representations get corrupted and can no longer be used by a network of the same architecture to productively classify images. To investigate this, after having trained the network on Task 2, we freeze parameters of stages 1 through $N$ to their post-task-2 values, reset the parameters of stages $N + 1$ onwards to their post-task-1 values, and then retrain those parameters on the Task 1 loss function. This is a natural, operational measure of the degree to which layer 1 through $N$ representations have been corrupted.

The results of this experiment, shown in Figure 14, tell a similar story to that of Figure **??** of the main text. In particular, we see even retraining the network with all but the last couple of layers frozen recovers nearly the pre-forgetting performance of the network.

Some of the performances in Figure 14 even outperform the pre-forgetting network performance, suggesting that network representations of intermediate layers may actually be improved by training on Task 2. Further suggestive results are shown in Figure 15, in which we retrain the full network

again on Task 1 after the training on Task 2. For ResNet and, to some degree, VGG, training on the second task clearly improves performance on the first.

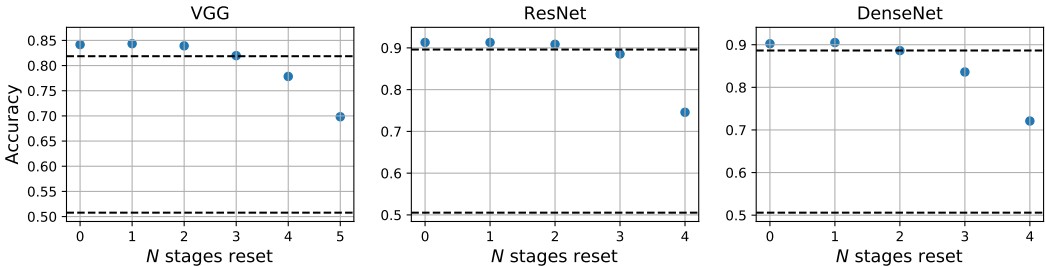

Figure 14: **Layer reset and retrain experiments.** Consistent with the observations in the freeze-training, layer reset, and CKA experiments, resetting and retraining only a few stages from the top is enough to recover the performance before forgetting.

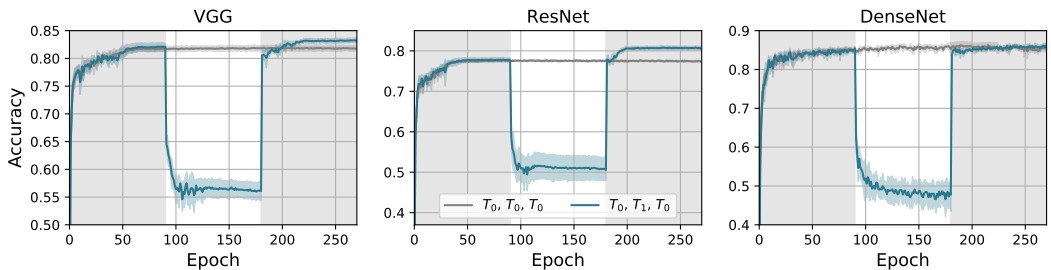

Figure 15: **Training on a different task can improve performance on the original task**. Networks are trained on two five-class splits of CIFAR-10. The gray curve shows the test accuracy on task 0 during training on solely that task for 270 epochs; the teal curve shows the task-0 test accuracy when training on task 0 for 90 epochs, then on task 1 for 90 epochs, and task 0 again for 90 epochs. While we see a sharp performance drop for the duration of training on task 1, the performance is quickly recovered once we begin training on task 0, and in fact surpasses the performance of the model only exposed to task-0 data. This effect is consistent across different splits of the original dataset. Of the architectures we studied, this effect is most pronounced for ResNet, but also exists to some degree for VGG and DenseNet models.

## B.4 Representation changes in the CIFAR-100 Distribution shift experiments

Here we show the change in representations due to training on distribution-shifted CIFAR-100 for the VGG (Figure 16) and DenseNet (Figure 17) networks, to accompany the ResNet results in the main text.

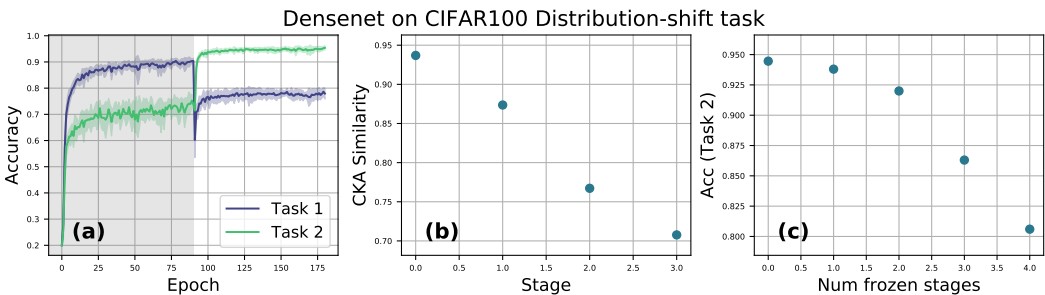

Figure 16: **Change in Densenet representations under CIFAR-100 distribution shift**.

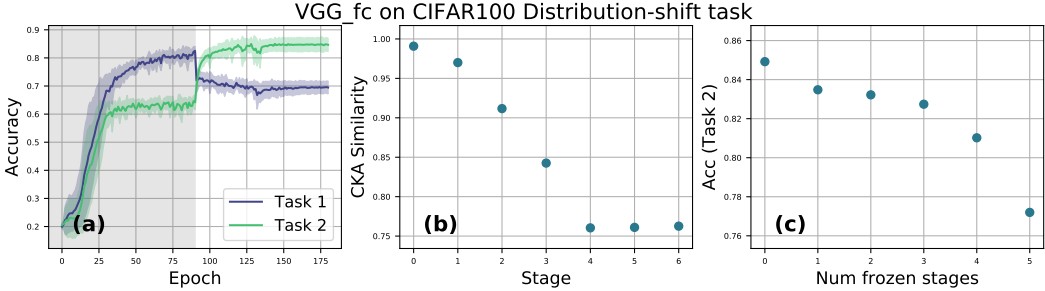

Figure 17: **Change in VGG representations under CIFAR-100 distribution shift**.

### B.5 ADDITIONAL EXPERIMENTS FOR SUBSPACE SIMILARITY OF MITIGATION METHODS

We include additional experiments on computing subspace similarities for different mitigation methods across different architectures. The results support the orthogonal storage of Task 1, Task 2 representations by replay, while EWC and SI enforce greater feature reuse in the higher layers.

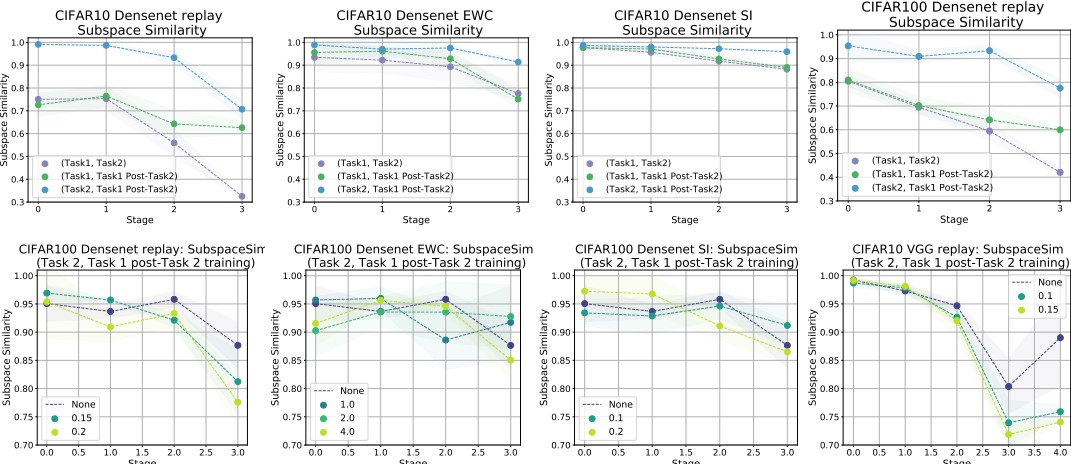

Figure 18: **Subspace similarity analysis reveals differences between different mitigation methods, with replay storing Task 1 and Task 2 representations in orthogonal subspaces, while EWC and SI promote feature reuse in the higher layers.** Additional results, compare to Figure 6 in the main text.

### B.6 LINEAR REGRESSION ON FORGOTTEN NETWORK ACTIVATIONS

To further probe the manner in which catastrophic forgetting affects the internal representation of networks, we perform the following experiment (in the split CIFAR-10 setting). We train a (multi-head) model on the first task for some time, resulting in model $M_1$, and then train on the second task for some time, resulting in model $M_2$. Due to forgetting, the model $M_2$ of course performs significantly worse than does $M_1$ on the first task. But, to probe how much information has been lost internally, we train a linear model to classify between the first-task categories using only the internal activations of $M_2$ on the first-task images. We measure both the amount of performance which we can recover using this linear regression, and the weight placed by the linear model on various stages of the network.

Model performances are given in Table 2. For all three architectures, the performance of the model before forgetting is naturally the highest, with a performance drop of up to forty percent due to forgetting. Remarkably, however, training a linear model on the internal post-forgetting activations recovers a substantial fraction of the performance loss: for ResNet and VGG, the linear model on post-forgetting activations is only a percent less accurate than the model before forgetting. As a control, we also train a linear model on the activations of networks at initialization (denoted the

|  | DenseNet | ResNet | VGG |
|---|---|---|---|
| Pre-forgetting accuracy | 0.869 | 0.7714 | 0.7970 |
| Post-forgetting accuracy | 0.4754 | 0.5924 | 0.5877 |
| Linear regression on activations | 0.802 | 0.7592 | 0.7896 |
| Linear regression on random lift | 0.447 | 0.4198 | 0.543 |

Table 2: **Performance of linear model trained on internal activations**.

*random lift* in the table). Because the number of activations is much higher than the number of pixels in a CIFAR-10 image, this control is necessary to ensure that the boost in performance we see is not simply due to performing a high-dimensional lift which renders the data linearly separable. As the table shows, however, this random lift performance is quite poor.

Figure 19 shows the weights placed by the linear model on the various internal activations of the network. We use the activations both before and after forgetting. Before forgetting, a linear model trained on the activations places most importance on the final layers, as expected; however, after forgetting, the model learns to use information stored in earlier-layer representations of the network. The reduced weighting of the final layers is consistent with the experiments we describe in the main text showing that deeper representations suffer the most from forgetting.

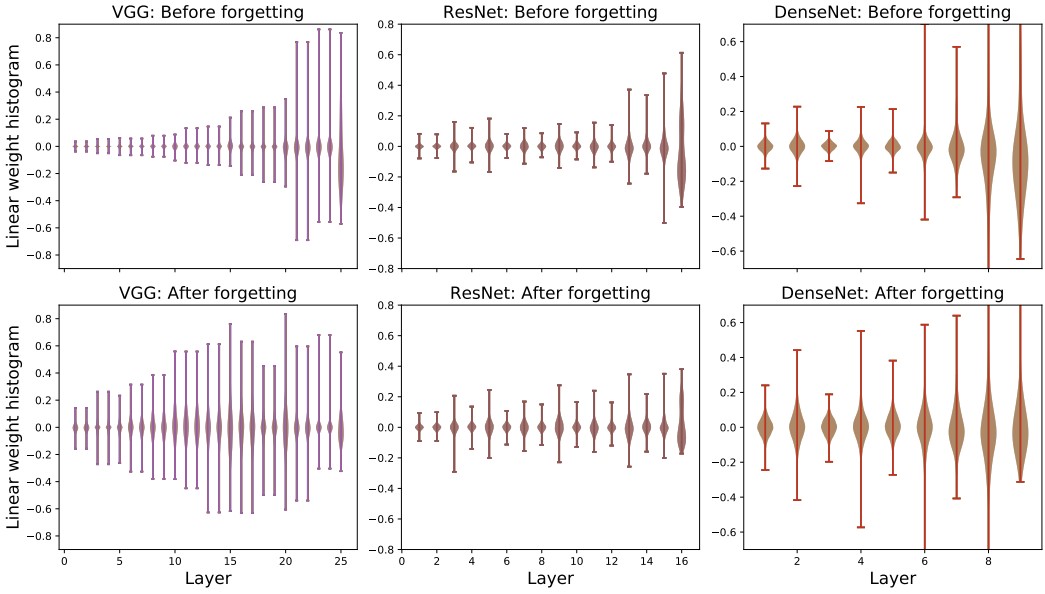

Figure 19: **Weights of linear models trained on internal activations.**

## B.7 CKA MEASUREMENTS ON MITIGATION METHODS

This section gives CKA similarity measurements to accompany those in the main text presented in figure 5. The main text figure showed results for ResNet; here we show corresponding results for all architectures on both split CIFAR10 and split CIFAR100 tasks, in figures 20 and 21, repsectively. These plots show that the results discussed in the main text apply broadly across architectures and datasets.

## B.8 OTHER CATEGORY IN RESNET AND DENSENET

In Figure 9 of the main text, we showed for the VGG architecture that adding an 'other' category (by sampling from all images in the dataset which are not in either of the main tasks) can mitigate forgetting by reducing the similarity between the model's representations of the task 1 data versus

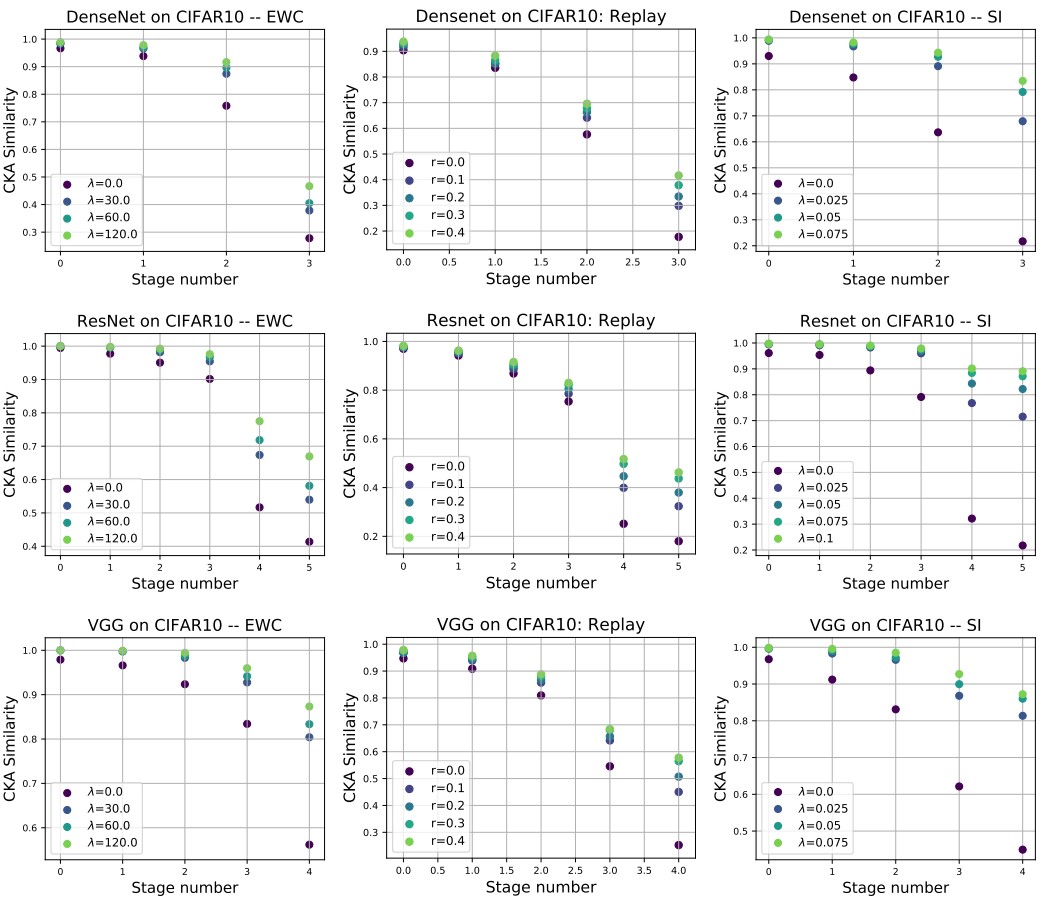

Figure 20: **CKA Analysis of mitigation methods on the split CIFAR10 task**

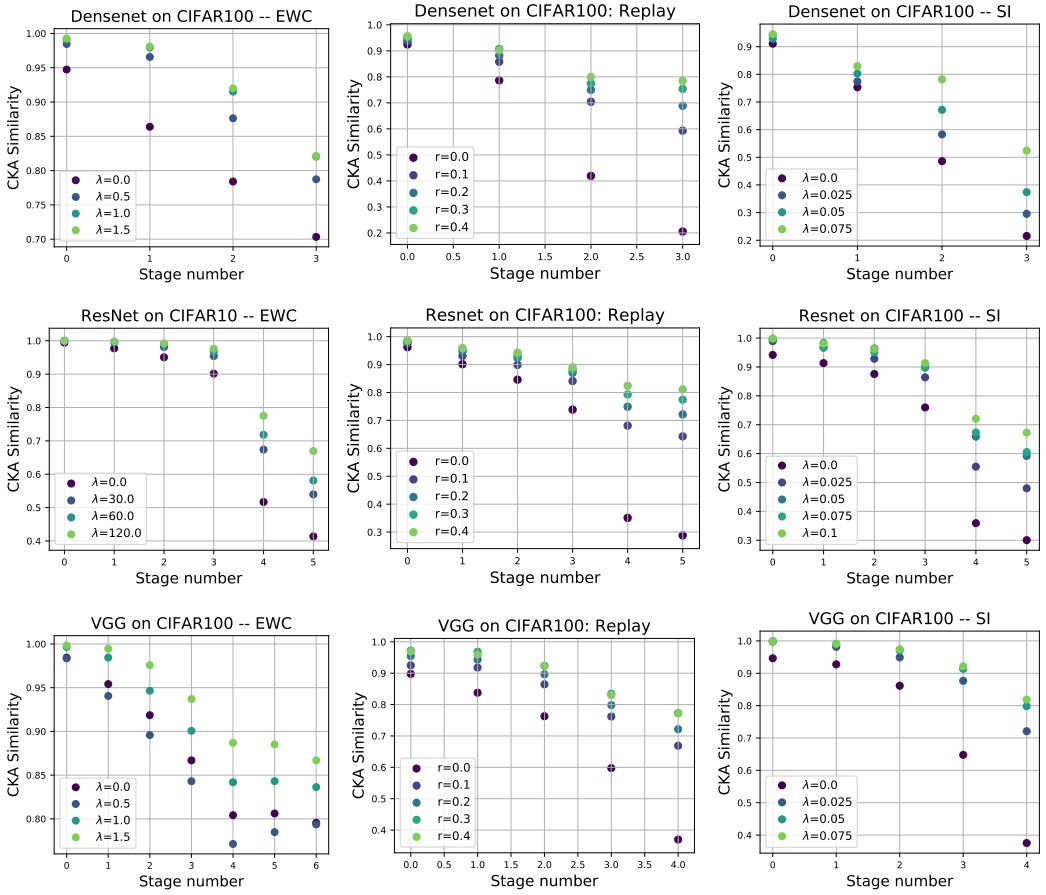

Figure 21: **CKA Analysis of mitigation methods on the split CIFAR100 task**

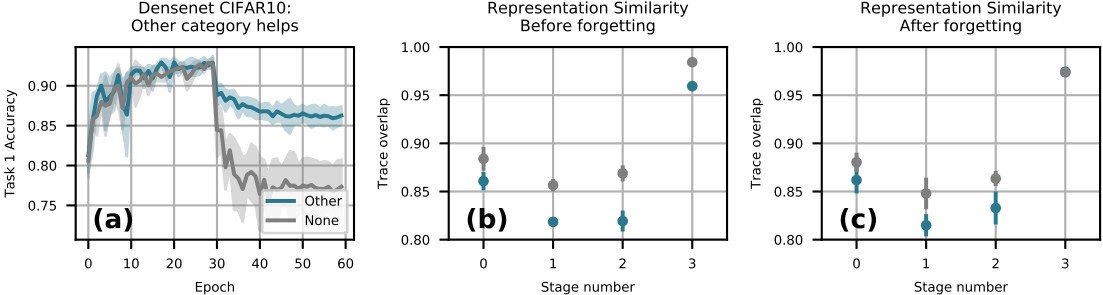

Figure 22: **Adding an 'other' category during initial training helps performance through orthogonalization: Densenet**. Model is trained initially on classifying two objects: airplane/automobile, and subsequently on two animals: deer/dog. When we add the 'other' category, it comprises a random sampling of images from all other classes. This reduces the similarity between the model's representations of each tasks' datapoints, leading to improved continual learning.

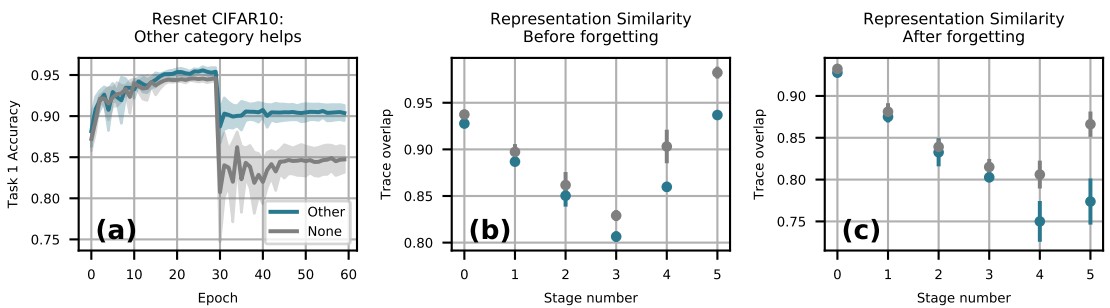

Figure 23: **Adding an 'other' category during initial training helps performance through orthogonalization: Resnet**. Model is trained initially on classifying two objects: airplane/automobile, and subsequently on two animals: deer/dog. When we add the 'other' category, it comprises a random sampling of images from all other classes. This reduces the similarity between the model's representations of each tasks' datapoints, leading to improved continual learning.

the task 2 data. As our analytic model suggested, increasing this dissimilarity should help forgetting. Figures 22 and 23 show corresponding plots for the Resnet and Densenet architectures.

## B.9 ADDITIONAL SEMANTIC EXPERIMENTS

In this section, comprising figures 24, 25, and 26, we present additional realizations of the semantic experiments presented in Section 6, Figure 8. In particular we present results for sequential binary classification tasks, four-class classification followed by two-class classification, and the CIFAR-100 distribution shift task. We find consistent results across different choices of objects and animals and across VGG, ResNet, and DenseNet architectures. In particular, we find that when models are encouraged to distinguish objects and animals, dissimilar categories lead to less forgetting. In contrast when models are trained with only a single semantic category, there is no pressure to distinguish objects from animals and similar categories lead to less forgetting. We observe one exception to the intuitive semantic groupings of categories. For VGG in Setup 2 (Figure 25) the truck category behaves similarly to animal classes in that the performance suffers more when the second task is animal classification then when it is object classification.

## C FROZEN-FEATURE MODEL: FURTHER DETAILS

Here we consider extensions and applications of the analytic model introduced in Section 6.2. Both the single and multi-head models relate the change in predictions during training on a second task to

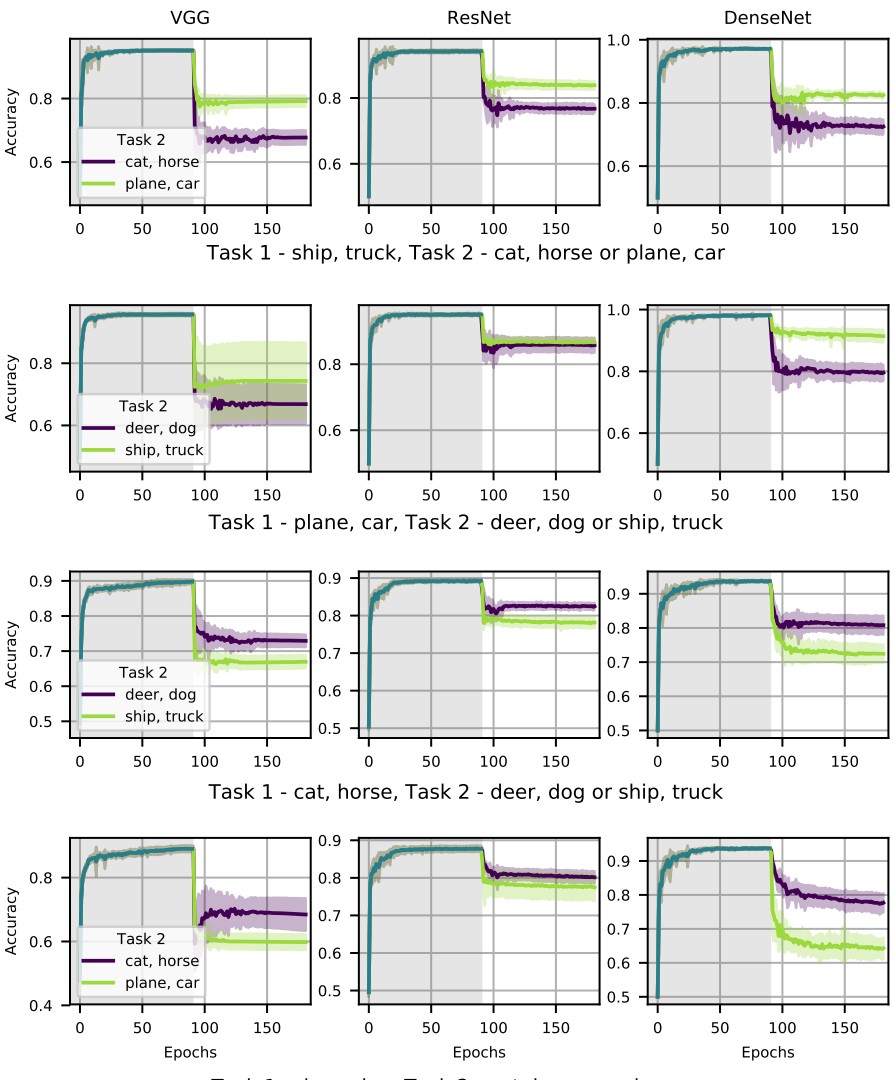

Figure 24: **Sequential binary classification tasks show consistent semantic structure.** We train on sequential binary subsets of CIFAR-10, distinguishing between two animals or two objects. We find that less forgetting occurs when the initial task is more similar to the second task.

the overlap between features, $\Theta(x, x') = \sum_\mu g_\mu(x) g_\mu(x')$, evaluated on the initial and second task data.[2]

**Multi-head model** In the main text we focused for simplicity on an analytic model of single head forgetting. Here we construct a solvable model for the multi-head setup. We consider a model

---

[2]This feature overlap matrix shows up frequently when studying linear models, where it is sometimes known as a reproducing kernel, or wide neural networks, where it is called the neural tangent kernel Jacot et al. (2018) and governs SGD evolution.

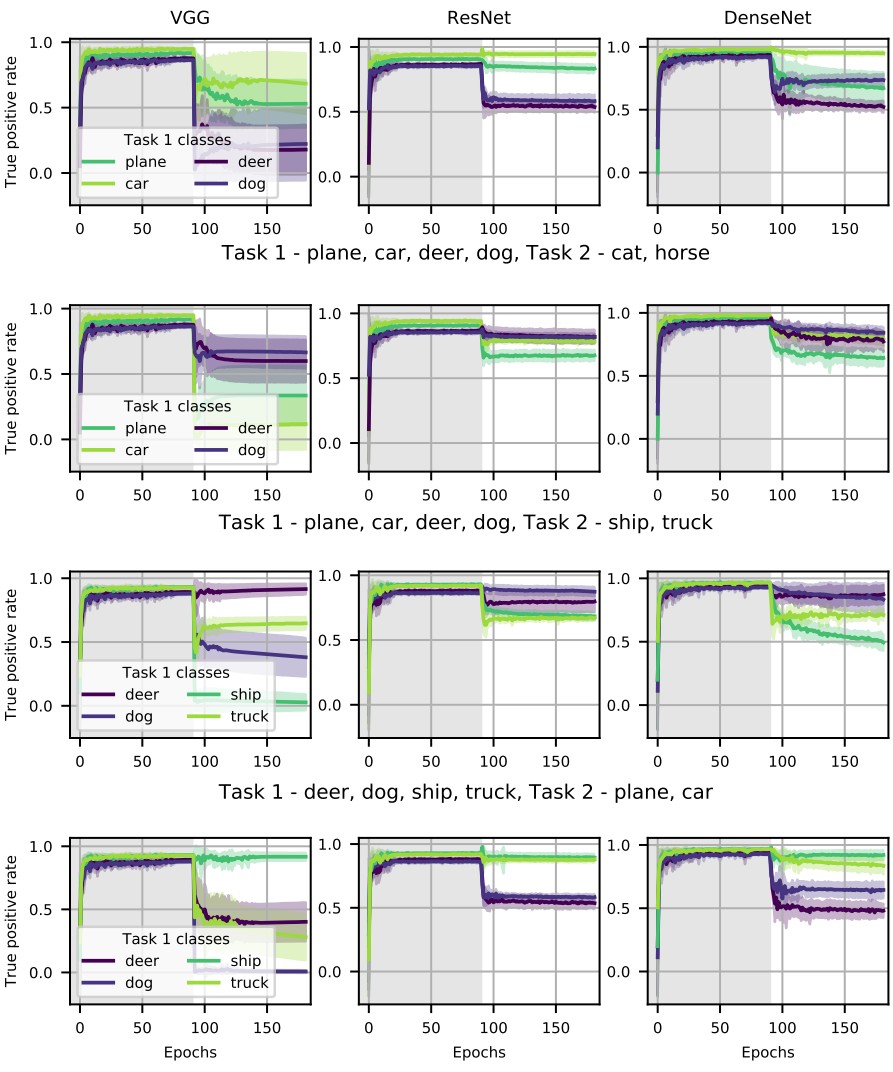

Figure 25: **Sequential four class and binary tasks show decreased forgetting for dissimilar tasks.** We train on sequential subsets of CIFAR-10, initially distinguishing between four classes (two animals and two objects) and then distinguishing between either two animals or two objects. We find that less forgetting occurs for categories that are dissimilar to those used in the second task.

consisting of non-linear features $g(w; x)$, a linear layer with weights, $\theta$, and a read out head $h^{(i)}$.

$$f^{(i)}(x) = \sum_{a,\mu} h_a^{(i)} \theta_{a\mu} g_\mu(w; x) \,. \tag{8}$$

Here $\mu$ runs over the feature dimensions, while $a$ runs over the output dimensions of our additional linear layer. For the initial task we use the head $h^{(1)}$, while for the second task, we swap the head and use $h^{(2)}$.

Again we consider a model trained without restrictions on Task 1 using head $h^{(1)}$. For Task 2, we first swap the head and then perform head only training on head $h^{(2)}$. After training the head, we

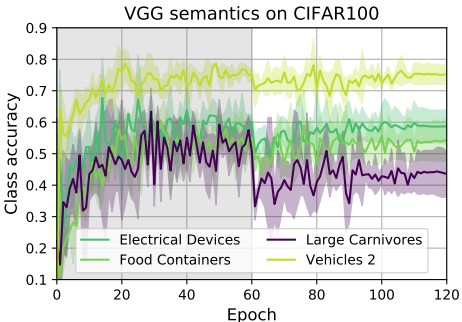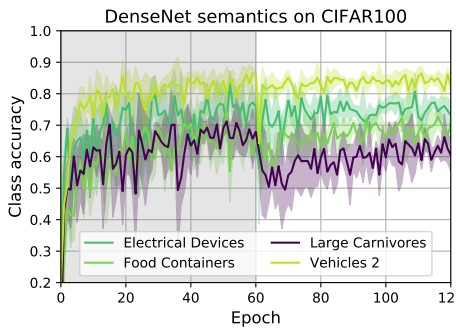

Figure 26: **Semantic structure of forgetting on the CIFAR-100 distribution-shfit task for VGG and DenseNet models**

|  | VGG | ResNet | DenseNet |
|---|---|---|---|
| No headfirst training | $0.147 \pm 0.051$ | $0.183 \pm 0.001$ | $0.162 \pm 0.008$ |
| 5 epochs headfirst training | $0.550 \pm 0.029$ | $0.673 \pm 0.010$ | $0.700 \pm 0.012$ |

Table 3: **Headfirst CKA similarity**. CKA similarity between the readout layer after headfirst training and after final training on the second task.

freeze the head, $h^{(2)}$, and the features $g(w; x) = g(\hat{w}; x)$, where $\hat{w}$ is the value of the feature weights after training on the initial task. This model is again inspired by our observation that forgetting is driven by changes in the latter network layers. The multi head model is additionally inspired by our observing that after head first training, the CKA similarity between the second task head before and after second task training is relatively high (see Table 3).

After training the second task head, we continue training the weights $\theta$. The model output evolves as

$$\Delta f_t^{(i)}(x) = -\eta \sum_{x', y' \in \mathcal{D}_{\text{train}}^{(2)}} \Theta(x, x') \frac{\partial L(f^{(j)}(x'), y')}{\partial f} h^{T(j)} h^{(i)} . \tag{9}$$

Again we find that if the representation overlap matrix $\Theta(x, x') = \sum_\mu g_\mu(\hat{w}; x) g_\mu(\hat{w}; x')$ is small between the features evaluated on the initial task and second task data then the predictions do not change significantly. If this overlap is zero, then the predictions are constant. In the multi-head setup we are considering here, we see that the change in predictions is further proportional to the similarity between the model heads, $h^{T(j)} h^{(i)}$.

Finally, we note that we have considered a final linear layer before the readout head for simplicity. One can instead include a ReLU non-linearity without changing the essential point, that the change in model output is governed by the overlap matrix, $\Theta$.

**Rotating representations in the analytic model.** We can use our analytic model (Section 6.2) to investigate how forgetting depends on representation similarity. We again consider sequential binary tasks (car vs plane) followed by (cat vs horse). We then tune the overlap of our initial task and final task features by explicitly rotating the frozen features for the second task to produce new features $g'(\theta; \hat{w}; x)$.

$$g'(\theta; \hat{w}; x) = R(\theta) g(\hat{w}; x) . \tag{10}$$

Here, if we have $P$ features, $R(\theta) \in \mathbb{R}^{P \times P}$ is a one-parameter family of rotation matrices designed such that $R(0) = \mathbf{1}$ and $g^T(\hat{w}; x) g'(\pi/2; \hat{w}; x')$ for $x \in X_{\text{test}}^{(1)}$ and $x' \in X_{\text{train}}^{(2)}$ is small. Explicitly, we take

$$R(\theta) = V^T \prod_{i=1}^{\lfloor P/2 \rfloor} r_{i, P-i}(\theta) V , \tag{11}$$

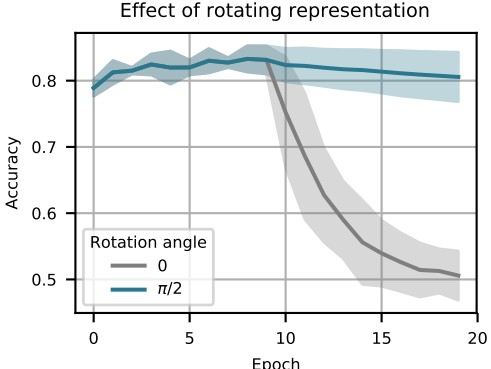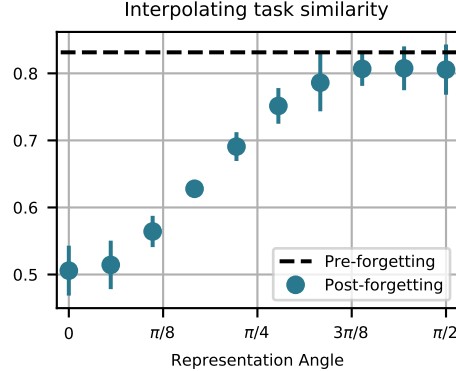

Figure 27: **Rotating representations shows diminished forgetting for dissimilar tasks**. Here we consider the performance of our frozen feature model on sequential binary CIFAR-10 tasks (Task 1: car vs plane, Task 2: cat vs horse). We use a model built from a two hidden-layer fully connected network with ReLU activations. We increase decrease representational similarity by explicitly rotating the second task features and find as predicted that this leads to diminished forgetting.

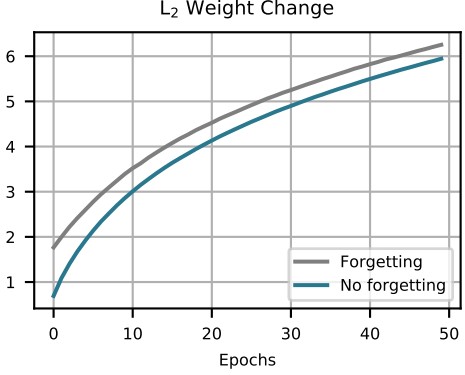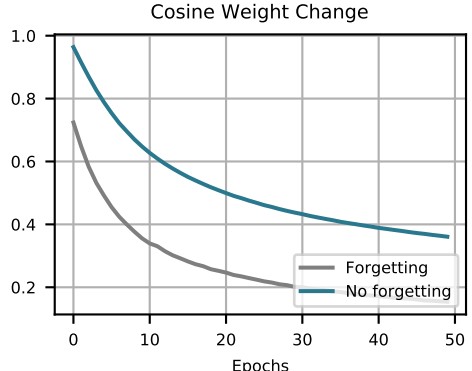

Figure 28: **Stabilizing final layer weights is not necessary to stop forgetting**. Here we look at the change in final layer weights during second task training for a model with severe forgetting (gray) and forgetting mitigated by rotating the second task representations (teal). We see that final layer weights become less and less similar in both cases. The setup is identical to Figure 27

where $r_{ij}(\theta)$ is the two dimensional rotation matrix between axes $i$ and $j$, and $V$ is the orthogonal matrix appearing in the singular value decomposition of the feature by data matrix on the initial task, $g(X_{\text{test}}^{(1)}) = UDV$. We find that explicitly enforcing task dissimilarity via a large rotation minimizes the effect of forgetting (Figure 27).

**Stopping forgetting without stabilizing weights** We saw in Section 4 that both EWC and replay buffers stabilize deeper network representations. As mentioned above, this need not be the case a priori. As an illustrative example, we construct a setup where sequential training leads to no forgetting despite a significant change to final layer weights. We consider the single head model of Section 6.2 where the Task 1 and Task 2 representations are completely orthogonal, $\Theta(x, x') = 0$ for $x \in X_{\text{test}}^{(1)}$ and $x' \in X_{\text{train}}^{(2)}$. In this case the Task 1 predictions remain constant throughout all of Task 2 training, despite significant changes to the final layer weights and predictions on the second task. In Figure 28 we present an example of this setup. The representations are again made orthogonal by explicitly rotating the second task features by the rotation matrix, $R(\pi/2)$ defined in Equation equation 11.

# D    ADDITIONAL EXPERIMENTS

This section presents two additional experiments which are slightly unrelated to the main thrust of the paper. We examine how the degree to which a network forgets is influenced by its width, and also how training the head independently of the rest of the network, in a multi-head setting, can mitigate forgetting.

## D.1    FORGETTING VERSUS NETWORK WIDTH

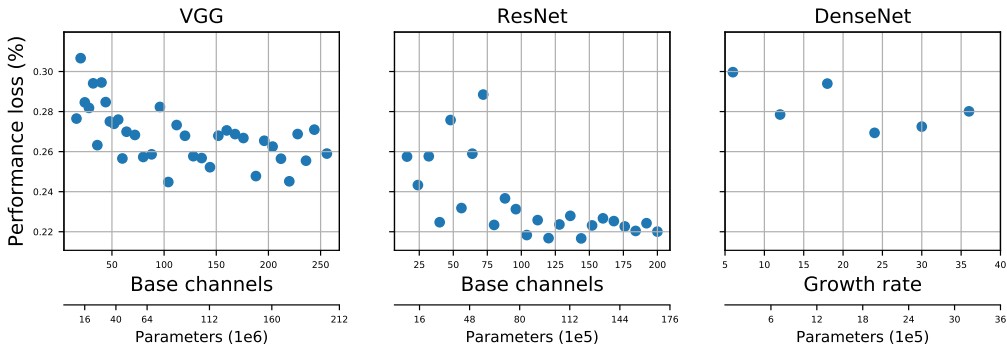

Figure 29: **Forgetting as a function of network width**. (a) VGG, (b) ResNet, and (c) DenseNet networks of varying widths were trained on two subsets of CIFAR-10 classes. We trained each of these (multi-head) networks for 30 epochs on each task with constant $\eta$, enough to achieve perfect training accuracy. Further training details can be found in the text. From the plots, which show the percent drop in accuracy (on task 1) due to training on task 2, the effect of width on the performance drop seems to be minimal, even sweeping across roughly a factor of ten in the number of model parameters.

Inspired by recent work of Gilboa & Gur-Ari (2019) showing that wider networks can learn better features than their narrower counterparts, we sought to determine whether wider networks also exhibit less catastrophic forgetting than narrower ones. We tested forgetting versus width for VGG, DenseNet, and ResNet networks on the split CIFAR-10 task in the multi-head setting. For VGG and ResNet, the width was varied by changing the number of channels in the initial convolutional layer and scaling the rest of the network layers concurrently while maintaining the network shape; for DenseNet, the width was varied by by changing the growth rate.

Surprisingly, though we found that the performances of the networks on each task improved as they grew wider, the amount of forgetting in each network, measured by the percent drop in accuracy (on task 1) due to training on task, exhibited a minimal dependence on network width. These results are shown in Figure 29.

## D.2    HEADFIRST TRAINING

When training multi-head networks on sequences of tasks, we found that the networks suffered a smaller performance drop on the original task, i.e. forgot less, if *only* the new task head was trained for a few epochs prior to training the full network. In Figure 30, this effect is shown for all three network architectures on the split CIFAR-10 task. Training only the head for up to five epochs seemed to improve the original-task performance without sacrificing performance on the new task. This suggests that coadaptation between the freshly-initialized head and the rest of the model contributes to forgetting when there is no headfirst training.

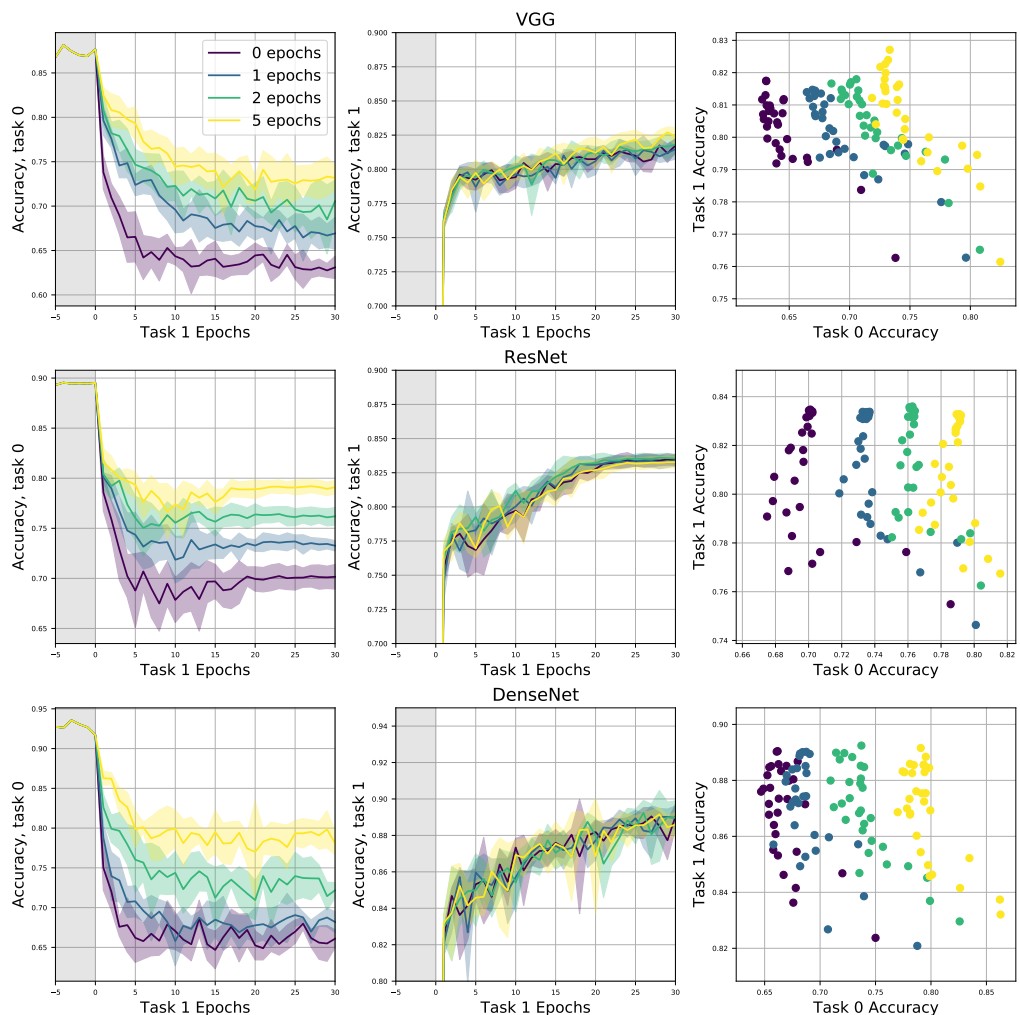

Figure 30: **Effect of training readout layer separately between task switches**. (a) VGG, (b) ResNet, and (c) DenseNet models were trained on two subsets of CIFAR-10 classes: task 0 (*dog*, *frog*, *horse*, *ship*, and *truck*), and task 1 (*airplane*, *automobile*, *bird*, *cat*, *deer*). When switching tasks, we first trained *only* the final classification layer (known as the *readout layer* or *head*) for a specified number of epochs (here 0, 1, 2, and 5) from a random Gaussian initialization. During this period, the rest of the model parameters are held fixed; after the head-only training, the entire model is trained as usual. Across all architectures, training only the head consistently improves the forgetting performance of the model (its accuracy on task 0) while having a negligible impact on the performance on the next task (task 1).

# E    FIGURES FOR ANONREVIEWER2

# F    IMAGENET RESULTS FOR TASK SEMANTICS

Figures corresponding to results tables in the main text.

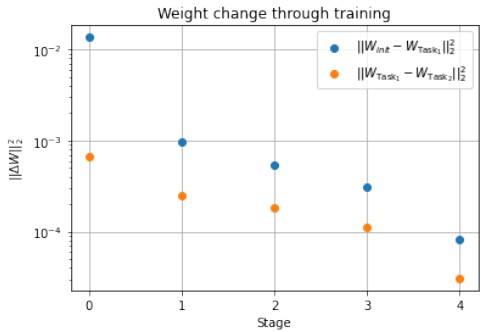 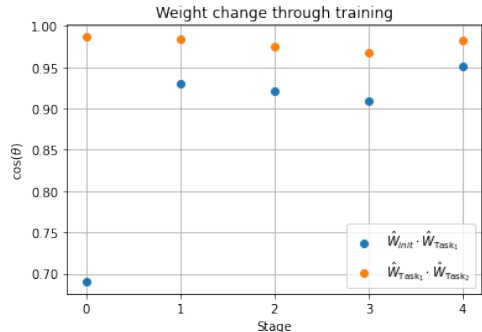

Figure 31: Stage-wise weight change between initial weights and weights after Task 1 (blue) and between weights after Task 1 and Task2 (orange) for a model trained sequentially on disjoint 5 class subsets of CIFAR10. (a) Mean $\ell_2$ distance. (b) Cosine similarity. We see no sign of significantly more change in last layer weights, and in fact see the reverse trend in $\ell_2$ distance.

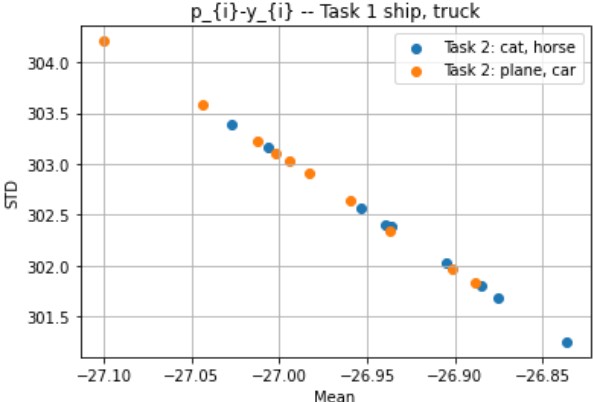

Figure 32: Mean and std deviation for the residual, $p_i - y_i$, for a model trained on *ship* v *truck* evaluated on *cat*, *horse* data (blue) and *plane*, *car* (orange). We see no clear separation between object and animal data. Each point represents a model trained with a different seed.

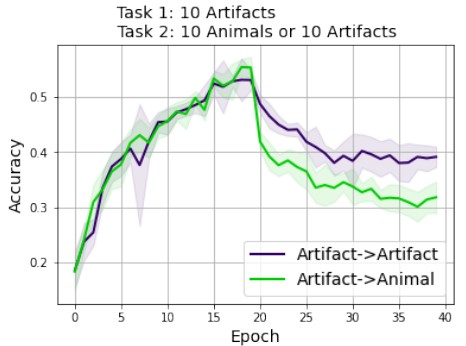 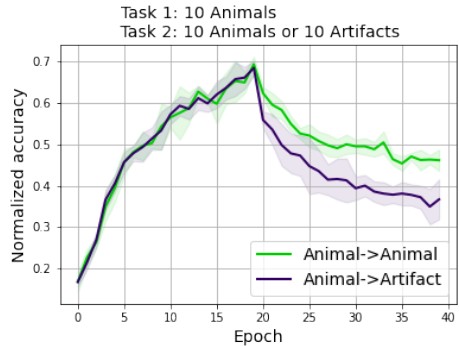

(a) Initial classification between 10 artifacts

(b) Initial classification between 10 animals

Figure 33: ImageNet Two-task 10 class sequences in which similar tasks cause less forgetting.

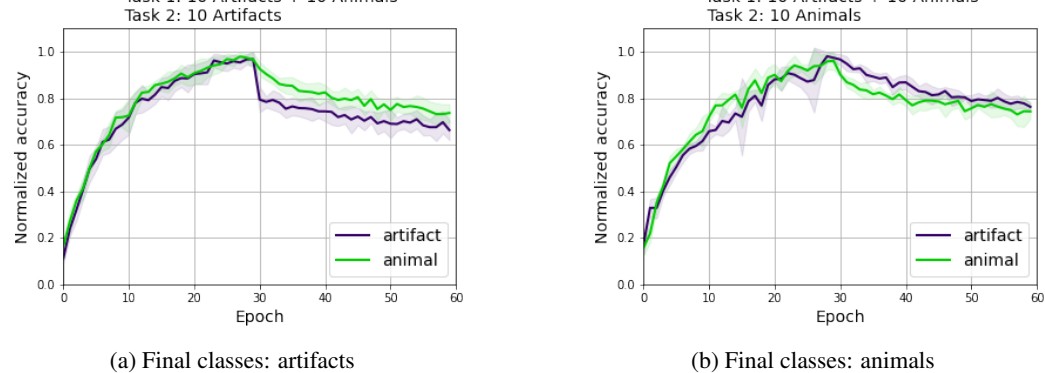

(a) Final classes: artifacts            (b) Final classes: animals

Figure 34: ImageNet Two-task 20-class followed by 10-class sequences in which similar categories are forgotten more.

