# OpenReview forum: "Anatomy of Catastrophic Forgetting: Hidden Representations and Task Semantics"
_ICLR.cc/2021/Conference — ICLR 2021 Poster_

### Official Review · AnonReviewer4 · 2020-10-27
**Interesting, but limited diversity of experiments and redundancy of some observations**

**Rating:** 7
**Confidence:** 4

**Review:**


This paper revisits the catastrophic forgetting problem that is common in multi-task learning. It concludes that the information encoded in the lower layers of the neural networks is more task-independent than higher layers, thus freezing the lower layer can reduce the risk of catastrophic forgetting. Another interesting finding is the semantic similarity between tasks can control the degree of forgetting. It also investigates several methods to mitigate this issue, with quantitive and analytical results to test their effectiveness.

Catastrophic forgetting is a common issue, especially in the areas where we need large pre-trained neural networks. Most of the previous methods measure the degree of catastrophic forgetting only by the task performance, and rarely looked into the parameters. This paper not only conducts experiments to test the task performance, but also the weights that the neural networks learned. Various methods are proposed, and the idea of "SubSpace Similarity" is very interesting, which studies whether the model predicts the two tasks with different subspaces.

The conclusion of section 6 is interesting, which claims that the datasets with dissimilar representations can reduce the forgetting, and the model could learn to utilize features of different subspaces for different tasks. This explains why sometimes semantically similar tasks will forget more.

However, the novelty of this paper is limited. It's already well-known that the lower layer of the neural networks captures more general knowledge and it's already good practice to only fine-tune the higher layers for transfer learning. E.g., in CV, it's common to use VGG only as a feature extractor. In NLP, fine-tuning high layers of the pre-trained transformer may yield better results. For example:  "Beto, Bentz, Becas: The Surprising Cross-Lingual Effectiveness of BERT" (https://www.aclweb.org/anthology/D19-1077.pdf)

The analysis of hidden layer representation is an application of the CKA method. The following subspace analysis is similar to section 6.4 of Kornblith et al 2019 but with CKA added. Section 5 is an analysis of existing mitigation methods and no new method is proposed. The statement that the replay method results in lower subspace similarity is interesting, but no explanation follows.

Some conclusions lack evidence. The results of this paper are based on CIFAR data with a few different settings, with conclusions supported by limited diversity of experiments. For example, to support the statement that representationally different tasks suffer less from forgetting, it only conducts two experiments ("other category" and "2v2 + 4v2") with three different datasets. The findings here would be more convincing with a broader range of tasks / data.

The presentation of the formulas could be improved. E.g, the denominator of eq(3) seems strange. The fonts of the first equation of section 4.2 is wrong, and the "x" above that equation should be replaced with \times. Some of the symbols should be defined before using, like the "w" at the beginning of section 6.2. Also, given that w is fixed, using symbol \hat{w} seems redundant. The section definition of Lemma 1 is not conventional because the curly brackets usually represent a set. The norm of equation (2) is undefined (I guess it's l2 norm because of Cauchy-Schwarz inequality).

Question:

In fig 3, you show that higher layers are disproportionately responsible for the forgetting, but the dissimilarity in lower layers will be propagated to higher layers. How do you minimize the effect of propagation?

Typo:
The definition of lambda in "Maximal forgetting at intermediate similarity" paragraph and the caption of Fig 9 are different.

---

> ### Author Response · Authors · 2020-11-21
> **Author Response to AnonReviewer4**
>
> We thank the reviewer for their time and feedback.
>
> **Additional experiments:** We have submitted a revised version of the paper that contains additional results on CelebA and ImageNet. Specifically:
> We have added CKA analysis of layer representations on CelebA and ImageNet, which also shows that lower layer representations are stable during sequential training, while higher layers change significantly.
>
> We repeat the subspace erasure and mitigation analysis on CelebA (Figure 7), finding that, like in CIFAR 10/100, higher layer representations of Task 1 undergo network subspace erasure during Task 2 training, and replay works by ensuring Task 1/Task 2 network subspaces are orthogonal.
>
> We study task semantics on ImageNet (Table 1), where, like CIFAR-10 and CIFAR-100, we observe that depending on the training setup, similar tasks can suffer greater forgetting than dissimilar tasks.
>
> **Layer representations and forgetting:** We agree that it is intuitive that lower layers perform feature reuse, but as far as we are aware, this is the first such analysis in the continual learning setting. More importantly, it is helpful to have the full picture of how continual learning affects network representations: lower layers performing feature reuse, higher layer layer representations changing, with subspace erasure of earlier tasks. This latter point also highlights an interesting difference between different mitigation strategies, and will be informative for future work on developing new mitigation methods.
>
> **Additional Task Semantics Results on ImageNet:** As mentioned above, we have included results on ImageNet for the task semantics section, where we again observe that representationally different tasks can suffer less forgetting. Note that the appendix (of the original submitted version) also contained semantics results on CIFAR100.
>
> **Figure 3, higher and lower layers:** Figure 3 measures the total change in layer representation before/after task 2 training, so is meant to also incorporate changes from earlier layers. But across all architectures/datasets, we see a significant drop in CKA in the final couple of layers, suggesting significant change is concentrated there. This is further supported by the layer freezing results (where earlier layers are frozen to stop the propagation of small changes), as well as the layer reset/retrain results in the Appendix.

---

### Official Review · AnonReviewer2 · 2020-10-27
**Interesting analysis but the conclusions seem obvious**

**Rating:** 6
**Confidence:** 5

**Review:**

**Summary**

The paper studies the role of (the depth of) hidden layers on catastrophic forgetting when a neural network is trained sequentially. For this, first, the paper analyses the similarity of the hidden representations through the centered kernel alignment (CKA) and subspace similarity measures. This analysis suggests that the deeper layers become more dissimilar when training is performed on a new task. Further, the subspace analysis suggests that sequential training forces the network representations of the earlier tasks to lie in the same subspace as that of the more recent task. The paper then studies the effect of different mitigation strategies for catastrophic forgetting and shows that for experience replay type approaches the performance is retained by keeping the subspaces of different tasks from overlapping from each other, whereas, in the regularization-type approaches the performance is retained by keeping the subspaces similar and resuing the previous feature mappings. Finally, the paper studies the effect of task similarity on catastrophic forgetting and shows that the intermediately similar tasks are the ones that are most prone to forgetting. The analysis is done on CIFAR-10 and a variant of CIFAR-100 datasets.

**Positives**

1- The experiments are done thoroughly and different hypotheses are explored in detail.


2- I quite like the task semantics part of the paper(Section 6). The authors develop a nice and somewhat novel argument in that section although there exists a simpler explanation which I will describe below.

**Negatives**

1- Meta comment: I can’t help but feel that the authors had it backward. Instead of looking at which layers contribute to more forgetting, it should be which layers forgetting effect the most. The latter lens changes the perspective of looking at the problem and most of the conclusions drawn in the paper seem obvious. Perhaps, then, most of my critique of the paper would be from that perspective.


2- **Deeper layers get more affected by the backprop**: Forgetting of previous tasks usually happens when the weights are changed from the optimal value of previous tasks. The change in weight is directly related to the norm of gradient received on a certain layer. In most deep networks, on image classification tasks this norm decreases as the gradient backpropagates to earlier layers. Hence, one sees that the weights of the earlier layers remain stable whereas the weights of deeper layers change quite a lot. So it is quite natural to observe that with training on a new task the representations of deeper layers change quite a bit. Note, I reach a similar conclusion as the authors but I don’t call it deeper layers causing more forgetting but rather forgetting causing deeper layers to change more. So to me, this contribution of the paper is stating the obvious.

3- **Sec 4.1**: The authors write that they observe that lower layers can reliably be frozen with very little impact on Task 2 accuracy. This again is obvious for natural image classification-based benchmarks (take fine-tuning for example). Perhaps, this conclusion will not hold for permuted MNIST benchmark where input distribution is quite different for each task.

4- **Task Semantics**: While this section is quite good and I enjoyed reading how the authors build the section but the puzzle that the authors describe at the outset can be explained more simply. Look at the gradient of the softmax-CE loss received at the logits. For a given image, it would be $p_i - y_i$, where $y_i$ is one-hot encoded (i.e.) 1 for true class and 0 for all other classes, and $i \in \\{1, \cdots, C\\}$. In Fig 7 (a, d), when binary classification is performed, the task 1 performance remains good when similar-looking objects are showed again for task 2 (and hence forgetting is less for similar tasks). More interestingly, Fig 7 (b), in a four-way classification task, when task 2 contains plane and car, the probability assigned to the logits of deer and dog would be very small (meaning smaller gradient and hence no change in the corresponding last layer weights), whereas the probability assigned to the logits of ship and truck would be higher which means higher gradient and more change in weights and more forgetting. A similar argument can be built for 7(e). It seems this is the artifact of the setup, single-/ multi-head, etc. Please correct me if I am wrong.

5- It seems that the paper is written hastily. There are a few typos.
*) Sec 3, line 2, we generally use
*) Sec 4.2, when defining X and Y, the dimension of the matrix should contain \times and not x.

---------------------------------------------------------------------------
**Post-rebuttal**

See my post-rebuttal response below. I have some remaining questions but the authors addressed some of my concerns. Therefore, I am increasing my score.

----------------------------------------------------------------------------

Looks like the post-rebuttal response is not available for the authors to see. I am copy-pasting it here.

I thank the authors for their detailed rebuttal.

**Deeper layers get more affected by backprop:** I appreciate the authors' response. The paper they referred to does not seem to have conclusive proof of which layers change more during the training. For example, Figure 1 in that paper shows that on FCNs, the latter layers change more. On CNNs, Figures 14, 15, the results are less conclusive. Anyhow, when I wrote the review the setup that I had in mind was a solution of the previous task as an initialization of the next task and not a random initialization and standard training -- a setup that had been the subject of the referred paper. In the former case (continual learning case), the solution of the previous task should inform on the next task and I expect the earlier layer weights to have small changes. However, what the authors of this paper showed in the revised draft (Figure 31) is quite interesting. They showed that, in the L2 sense, the weights of the earlier layers change more than that of the later layers. This shows that despite the large weight changes in the earlier layers their representations don't change much (as measured by the CKA). Am I understanding that correctly? Could I request the authors to verify this observation on ResNets (I believe Figure 31 is for VGG), and perhaps, other datasets as well?

**Sec 4.1**: Regarding freezing the earlier layers and not seeing any performance degradation on the subsequent tasks, I asked whether this would also hold for setups where input distribution changes (e.g.) Permuted MNIST. Could the authors please address this and add the experiment to the paper?

My other concerns are addressed, therefore, I am increasing my score.

---

> ### Author Response · Authors · 2020-11-21
> **Author Response to AnonReviewer2**
>
> We thank the reviewer for their time and feedback, though we respectfully disagree with some of the points raised in the review (e.g. deeper layers affected more by backprop), which have been shown to be false by other work.
>
> **Additional experiments:** We have submitted a revised version of the paper that contains additional results on CelebA and ImageNet. Specifically:
> We have added CKA analysis of layer representations on CelebA and ImageNet, which also shows that lower layer representations are stable during sequential training, while higher layers change significantly.
>
> We repeat the subspace erasure and mitigation analysis on CelebA (Figure 7), finding that, like in CIFAR 10/100, higher layer representations of Task 1 undergo network subspace erasure during Task 2 training, and replay works by ensuring Task 1/Task 2 network subspaces are orthogonal.
>
> We study task semantics on ImageNet (Table 1), where, like CIFAR-10 and CIFAR-100, we observe that depending on the training setup, similar tasks can suffer greater forgetting than dissimilar tasks.
>
> **Deeper layers get more affected by backprop:** This statement is incorrect. In general, the norms of the gradients of different layers will depend on many factors such as the scale of initialization, use of batch norm, etc. Multiple papers (one recent example is https://arxiv.org/abs/1902.01996 ) have studied how much different layer parameters change during training on standard image classification tasks, where it is typically found that the lowest convolutional layers change significantly through the training process (and are key to the task).
>
> To further refute this being the case, we performed measurements on the degree of weight change in our setting. In supplementary section E Figure 33 we present measurements of the mean l2 norm and cosine distance of the change in weights between initialization and the end of Task 1 and between the end of Task 1 and the end of Task 2. We find no evidence that the latter layer weights change significantly more than earlier layers and indeed find the opposite trend for the l2 change.
>
> **Sec 4.1:** It is indeed intuitive that lower layer features might be reused, although as far as we are aware, this has not been studied in a continual learning setting. But most importantly, the results of Sections 4.1, 4.2 and 4.3 are meant to be taken in conjunction to give a full, precise picture of what happens during catastrophic forgetting: we see that lower layers don’t change much through sequential training, higher layer representations change significantly, and experience subspace erasure of the earlier task.
>
> **Task Semantics:** We would be enthusiastic about an even simpler argument for the semantic behavior we see in our paper. Unfortunately, we do not see the argument directly from the logit difference, however. Nonetheless we attempted to investigate this proposal quantitatively and have included some additional experiments. In particular in the context of sequential two way classification, we have trained a network on binary ship, truck classification, as in figure 7a and evaluated the residual, p_{i} - y_{i} for second task data with both animals (cat, horse) and objects (plane car). We see no significant difference in the residuals between these two datasets. See figure 34 in supplementary section E for the means and variances.
>
> In light of these clarifications, we respectfully ask that you consider increasing your score to support the acceptance of this paper.

---

### Official Review · AnonReviewer3 · 2020-10-27
**Collection of empirical observations about image representation and continual learning for which it is unclear how broadly they apply.**

**Rating:** 6
**Confidence:** 4

**Review:**

In this paper, the authors provide an empirical investigation that the outer layers of a neural network are more responsible for the catastrophic forgetting effect than inner layers.  This is established by demonstrating graceful performance decays in training a second CIFAR classification task when freezing inner layers after training a first task.  Additionally, in sequential training, various notions of similarity reveal that outer layers change more.  The paper reveals that continual learning techniques (EWC, SI, a replay method) mitigates the change in outer layer image representations, as per these metrics.  The paper reveals that some methods cause outer layer features to live in orthogonal subspaces to those of previous tasks, and other methods cause reuse of existing outer layer features from previous tasks.  The authors present an additional observation that sometimes similar sequential tasks results in less forgetting and sometimes it results in more forgetting.  They reveal experiments that show that in a mixup dataset, intermediate levels of task similarity lead to maximal forgetting.

The paper has a reasonably thorough set of random experiments exploring the effect of Continual Learning methods on feature representations.  That said, it is not clear how general these observations are.  For example in the EWC Paper (Kirkpatrick et al. 2017) it was observed that permutation tasks resulted in significant changes to inner layer representations (as measured by Fisher information) relative to outer layers.  At the very least this paper should reconcile or comment on this difference in observation with the presented results.  Ultimately, it is not clear to this reviewer what to take away from this paper.  The paper would be stronger if it was accompanied with an algorithmic improvement, inspired by the present observations, that improves on the state of the art.  That said, the reviewer suspects that the observations in this paper might reasonably lead to such an advance in subsequent work by the same or different authors.

Minor comment:

Sec 4.2: Is the SubspaceSim definition correct?  The numerator is a quartic, but the denominator is a quadratic.

---

> ### Author Response · Authors · 2020-11-21
> **Author Response to AnonReviewer3**
>
> Thank you for your comments and feedback.
>
> **Additional experiments:** We have submitted a revised version of the paper that contains additional results on CelebA and ImageNet. Specifically:
> We have added CKA analysis of layer representations on CelebA and ImageNet, which also shows that lower layer representations are stable during sequential training, while higher layers change significantly.
> We repeat the subspace erasure and mitigation analysis on CelebA (Figure 7), finding that, like in CIFAR 10/100, higher layer representations of Task 1 undergo network subspace erasure during Task 2 training, and replay works by ensuring Task 1/Task 2 network subspaces are orthogonal.
> We study task semantics on ImageNet (Table 1), where, like CIFAR-10 and CIFAR-100, we observe that depending on the training setup, similar tasks can suffer greater forgetting than dissimilar tasks.
>
> **Permutation tasks in Kirkpatrick et al 2017:** Permutation tasks like in Kirkpatrick et al, where the pixels in an MNIST digit are scrambled do not allow for feature reuse (the underlying attributes of the image, e.g. curves, edges, are all destroyed), and hence it would be expected that the lower layers may change in such a task, as features cannot be reused between Task 1 and Task 2. In fact, in Kirkpatrick et al, they observe that when the permutation size is smaller (not all pixels are scrambled), there is significantly higher overlap in the lower layers in Fisher information, as some feature reuse can be utilised.
>
> In this paper, we chose to focus on more natural forms of distribution shift (different categories, variation in inputs), as these are more representative of the typical ways catastrophic forgetting arises in practice.
>
> **Paper takeaways:** The paper offers the following takeaways:
> 1. Catastrophic forgetting arises due to large changes in the higher layer representations of neural networks. Specifically, during sequential training, lower layer representations remain stable, with features being reused on subsequent tasks. Higher layer representations change significantly however, with the network subspaces of earlier tasks being erased by later tasks.
>
> 2. Mitigation methods work to stabilize the representation changes in higher layers, but while regularization based mitigation (EWC, SI), achieves this by increasing feature reuse across all layers, replay stores different tasks in orthogonal network subspaces (seen by a decrease in higher layer subspace similarity between tasks)
>
> 3. Investigating how similarity between sequential tasks affects forgetting shows that surprisingly, in some settings tasks being more similar results in greater forgetting while in other settings, dissimilar tasks result in greater forgetting.
> We formalize this mathematically, showing that the intermediate similarity between tasks results in the greatest forgetting. From this, we also derive a measure of task similarity, trace overlap, and verify these results empirically.
> Informed by these insights, we study adding an “other category”, when training on the first task as a mitigation method. This encourages increased orthogonality of different task representations and reduces forgetting. We leave a more detailed study of the applications of this mitigation method to future work.
>
> **Subspace dim definition:** The definition is correct. This is a result of the fact that Vk and Uk are constructed from orthonormal basis vectors, so ||V_{k}||_{F}=||V_{k}^{T}V_{k}||_{F} = \sqrt{k}. The denominator just gives a factor of k. To avoid this confusion, we have now changed the notation in the text.
>
> **Summary:** We hope that these responses have helped address some of the questions about the paper, and respectfully request that you consider increasing your score to support the acceptance of this paper.

---

### Official Review · AnonReviewer1 · 2020-10-28
**Official Blind Review #1**

**Rating:** 7
**Confidence:** 3

**Review:**

##########################################################################

Summary:

The paper investigates catastrophic forgetting phenomena, where machine learnings model trained on a new domain suffer from performance losses on older trained domains. The paper first finds that deeper layers are disproportionately responsible for forgetting. Next, it analyzes the effects of mitigation methods on higher layers and suggests that they stabilize higher layer representations by either enforcing feature re-use or storing tasks in orthogonal subspaces. Lastly, using analytic models, the paper investigates the connection between forgetting and task semantics and shows that the intermediate similarity in sequential tasks leads to maximal forgetting. The paper empirically demonstrates the above results on CIFAR10 and CIFAR100.

##########################################################################

Pros:
1. The layer-wise analysis of catastrophic forgetting and investigation of different mitigating forgetting methods are interesting, and the work is certainly very relevant to this venue.
2. The motivation of the paper is clear, and the empirical results are convincing.
3. The writing is clear and straightforward.
4. The results can potentially help to suggest new approaches for developing and measuring mitigation methods.
5. The related work section is sufficient.

##########################################################################

Cons:
1. The paper misses important discussions that I describe in the section below.
2. Although the empirical findings seem useful, it would have been nicer to propose some new mitigation methods from these insights.
3. The experiments are only done on image classification tasks with CIFAR10 and CIFAR100

##########################################################################

Questions:
1. What does "N frozen stages" represent in Figure 2? Does it correspond to freezing N lower layers (blocks)? What happens if only the higher layers are frozen during training? The conclusion that higher layers contribute to catastrophic forgetting is not evident from the paper's experiments in section 4.1.
2. In figure 6, how does subspace similarity differ from one that does not use any mitigation methods? It is challenging to compare the results from figure 5 directly.
3. How is the task similarity defined? I assume it is the similarity of the before-output layer activations?
4. What are representation similarity before/after forgetting in figure 8 (b) and (c)?
5. The paper does not discuss the potential reasons why domains with intermediate similarity have maximum forgetting. Does it relate to findings in section 6.3 (re-use and orthogonality)? Additionally, what would happen if different mitigation methods are used?
6. What property (re-use vs. orthogonality) is advantageous in designing mitigation methods? It would be helpful to have this discussion in the paper.

Minor notes:
* All titles in the figures are relatively small and hard to read. It would be helpful if important parts are highlighted in the title.
* On page 2, "We generally uses m = 2" --> "We generally use m = 2"
* On page 2, "Models" --> "Models:"
* It will be more apparent if the equation on page 3 is defined as CKA(X, Y) = ... Also, the equation numbers are missing.
* On page 3, \mathbb{R}^{n \times p_1} instead of \mathbb{R}^{n x p_1}. Also, shouldn't p_1 = p_2?
* In page 7 "in Figure 7a and7d" --> "in Figure 7a and 7d"
* In page 7, is \hat{w} representing frozen weight?
* On page 7, \eta and t are not defined.
* On page 7, "puzzle in Figure  7" --> "puzzle in Figure 7"; two spaces.
* There are some \citep and \citet confusions.
* In Lemma 1, the notations (e.g. norm of the set) look awkward to me.

---

> ### Author Response · Authors · 2020-11-21
> **Author response to Reviewer 1 Part 1**
>
> We thank the reviewer for their positive feedback and comments!
>
> **Additional experiments:** We have submitted a revised version of the paper that contains additional results on CelebA and ImageNet. Specifically:
> We have added CKA analysis of layer representations on CelebA and ImageNet, which also shows that lower layer representations are stable during sequential training, while higher layers change significantly.
> We repeat the subspace erasure and mitigation analysis on CelebA (Figure 7), finding that, like in CIFAR 10/100, higher layer representations of Task 1 undergo network subspace erasure during Task 2 training, and replay works by ensuring Task 1/Task 2 network subspaces are orthogonal.
> We study task semantics on ImageNet (Table 1), where, like CIFAR-10 and CIFAR-100, we observe that depending on the training setup, similar tasks can suffer greater forgetting than dissimilar tasks.
>
> **New mitigation method:** Our focus in this paper was to better understand the phenomenon of catastrophic forgetting, and through thorough experiments across multiple datasets/architectures/experimental configurations, which were also mathematically analyzed, we were able to gain many robust insights (described above.) Informed by this, we also introduced the “other category”, which is a simple, novel mitigation method. We leave further exploration of this to future work.
>
> **N frozen stages and higher layers and forgetting:** “N frozen stages” does indeed correspond to freezing the first N blocks (stages).
>
> This result demonstrates that even with forced feature reuse in the lower layers, high task 2 accuracy can be achieved. Namely, the lower layers learn general features on Task 1 that are capable of being reused as is on Task 2. Next, the CKA results in Figure 3 show that these lower layer general features are indeed reused on Task 2: we observe high CKA similarity between lower layer representations before and after Task 2 training, while higher layer representations show low CKA similarity, and are changing significantly during training. Taken together, these results strongly suggest that change in higher layers is responsible for catastrophic forgetting.
>
> To further support this conclusion, in the appendix we include (i) layer reset (Figure 11 in original supplementary) and (ii) layer retrain (Figure 13 in original supplementary) experiments. (Note these are also referred to and summarized in the text in Section 4.2)  In layer resets, we rewind the layer parameter values after training on Task 2 back to their values after Task 1 training. We observe that resetting a few of the highest layer parameters alone results in an immediate (no training), significant increase in accuracy on Task 1, while resetting lower layer parameters alone has minimal effect on Task 1 accuracy. The layer retrain experiments are very similar to your suggested comment of freezing the higher layers: in this experiment the lowest N stages are frozen after training on Task 2, while the remaining layers are set to their values after Task 1. We evaluate this on Task 1, finding that adapting the highest layers to Task 1 in this way again significantly improves Task 1 accuracy.
>
> Taken together, all of these results clearly demonstrate change in higher layers being the main contributor to catastrophic forgetting.
>
> **Figure 6 subspace similarity and no mitigation:** The results in Figure 6 can be directly compared to Figure 4. Additionally, the bottom row of Figure 6 includes results with no mitigation for ease of comparison. We have added text to emphasise the main points, specifically:
> Comparing the top row of Figure 6 to Figure 4 (the same plot without mitigation), we see that mitigation methods increase the subspace similarity of (Task 1, Task 1 Post-Task 2) and (Task 1, Task 2) representations in earlier/middling layers (more feature reuse).
> Most interestingly, comparing the different mitigation methods across top row Figure 6, we see that while EWC and SI maintain high subspace similarity of (Task 2, Task 1 Post-Task 2) even in the highest layers, replay significantly decreases this value in the highest layers. This suggests that replay performs mitigation through use of orthogonal subspaces, while EWC/SI encourage feature reuse even in the highest layers.
> In the bottom row of Figure 6, we show subspace similarity of (Task 2, Task 1 Post-Task 2) for varying strengths of the different mitigation methods. We observe a clear decrease in subspace similarity in higher layers even when a little replay is used (again supporting orthogonal subspaces), while EWC/SI maintain or increase this (feature reuse even in higher layers.)

---

> > ### Author Response · Authors · 2020-11-21
> > **Author Response to Reviewer 1 Part 2**
> >
> > **How is task similarity defined:** We use the trace overlap defined in equation (3), and using the before-output layer activations, as a concrete measure of task similarity.
> >
> > **Figure 8 (b) and (c):** We apologize if this was confusing. Again here we use the trace overlap (as indicated on the y axis) to quantify similarity.
> >
> > **Domains with intermediate similarity have maximal forgetting:** The reason domains with intermediate similarity have maximal forgetting does indeed touch on both feature reuse and orthogonality, which we formalize in Lemma 1 (and the derived trace overlap metric). At a high level, because the domains have some dissimilarity, features cannot be entirely reused. At the same time, the domains are similar enough that they are not stored in orthogonal subspaces in the network, meaning that sequential training (with no mitigation) results in interference between the network subspaces. Together, this causes maximal forgetting.
> >
> > **Reuse vs orthogonality in mitigation:** the results of Section 5 show that it is not “reuse vs orthogonality” but reuse and orthogonality --- implemented correctly, each is a successful, distinct approach to performing mitigation. Informed by our findings that regularization based mitigation approaches primarily rely on feature reuse, and replay mitigation approaches rely on orthogonality, future work can explore combining both strategies.
> >
> > In light of these responses, we respectfully request that you consider increasing your score and confidence to support the acceptance of this paper.

---

> > > ### Comment · AnonReviewer1 · 2020-11-24
> > > **Response to Authors**
> > >
> > > I thank the authors for the detailed reply. I carefully read the authors' responses (updated manuscript) and other reviewers' feedback.
> > >
> > > **Additional experiments**: Thank you for including the additional experiments. I agree that these experiments confirm the claims made in the papers.
> > >
> > > **N frozen stages and higher layers and forgetting**: I agree that the overall experiments demonstrate that the higher layers contribute to catastrophic forgetting. However, experiments in section 4.1 are not sufficient enough to claim "the possibility of lower-layer features being reused between both tasks, with higher layers being the main contributor to catastrophic forgetting." I would be interested in the effect of freezing the higher layers as well. Nevertheless, as I mentioned above, I acknowledge that the overall experiments support that the higher layers contribute to catastrophic forgetting more.
> > >
> > > **Others**: The revised version of the paper addresses my concerns and some of the other reviewers' concerns. I updated the score accordingly. However, I am not able to increase my confidence.

---

> > > > ### Author Response · Authors · 2020-11-25
> > > > **Response to Reviewer1**
> > > >
> > > > Thank you for your response! We can definitely include an experiment freezing higher layers in the next revision! Though based on the current results, we expect that the outcome of this experiment will be similar to the layer reset/retrain experiments, and (as you have also described) support the finding that changes in higher layers representations are a primary contributor to forgetting.

---

### Decision · Program_Chairs · 2021-01-07
**Final Decision**

**Decision:**

Accept (Poster)

**Comment:**

This paper studies how layer-wise representation and task semantics affect catastrophic forgetting in continual learning. It presents two findings: 1. the higher layers contribute more to forgetting than lower layers, 2. intermediate-level similarity between tasks causes the maximal forgetting. It also indicates that existing methods employ either feature reuse or orthogonality to mitigate forgetting.

Pros:
- The layer-wise analysis of catastrophic forgetting and investigation of different mitigating forgetting methods are important and interesting.
- The paper is well-motivated and well-written.
- The results can potentially help to suggest new approaches for developing and measuring mitigation methods.

Cons before rebuttal:
- The paper misses discussion on and takeaways from the findings.
- How general are the findings? There is a different observation by Kirkpatrick et al. 2017.
- Limited diversity of experiments, because the experiments are only done on image classification tasks with CIFAR10 and CIFAR100.

The authors conducted more experiments and updated the paper with added explanations and results. The reviewers found the new evidence and arguments in the rebuttal to be convincing and the authors addressed most concerns.

In summary, the findings from this paper will help researchers better understanding and addressing catastrophic forgetting, and will be of interest to the community.
Hence, I recommend acceptance.